# Selective sorting and destruction of mitochondrial membrane proteins in aged yeast

Adam L Hughes[1,2]*, Casey E Hughes[2], Kiersten A Henderson[1], Nina Yazvenko[1], Daniel E Gottschling[1]*†

[1]Division of Basic Sciences, Fred Hutchinson Cancer Research Center, Seattle, United States; [2]Department of Biochemistry, University of Utah School of Medicine, Salt Lake City, United States

**Abstract** Mitochondrial dysfunction is a hallmark of aging, and underlies the development of many diseases. Cells maintain mitochondrial homeostasis through a number of pathways that remodel the mitochondrial proteome or alter mitochondrial content during times of stress or metabolic adaptation. Here, using yeast as a model system, we identify a new mitochondrial degradation system that remodels the mitochondrial proteome of aged cells. Unlike many common mitochondrial degradation pathways, this system selectively removes a subset of membrane proteins from the mitochondrial inner and outer membranes, while leaving the remainder of the organelle intact. Selective removal of preexisting proteins is achieved by sorting into a mitochondrial-derived compartment, or MDC, followed by release through mitochondrial fission and elimination by autophagy. Formation of MDCs requires the import receptors Tom70/71, and failure to form these structures exacerbates preexisting mitochondrial dysfunction, suggesting that the MDC pathway provides protection to mitochondria in times of stress.

*For correspondence: hughes@biochem.utah.edu (ALH); dang@calicolabs.com (DEG)

Present address: †Calico Life Sciences, South San Francisco, United States

Competing interests: The authors declare that no competing interests exist.

## Introduction

Mitochondria play a central role in cellular metabolism. Metabolic pathways that occur within mitochondria include the TCA cycle, oxidative phosphorylation, amino acid metabolism, and biosynthesis of lipids, heme, and iron-sulfur clusters (*Rutter and Hughes, 2015*). Because so many vital processes take place within mitochondria, cells have evolved multiple mechanisms to maintain mitochondrial homeostasis under the diverse array of physiological states that a cell can exist in. These mechanisms are important for maintaining mitochondrial function during development, cellular specialization, aging and environmental challenges of nutrient availability or toxic stresses. Failure to maintain mitochondrial homeostasis under these conditions contributes to the development of numerous age-associated and metabolic disorders (*Nunnari and Suomalainen, 2012*).

Many of the adaptive responses made by mitochondria are manifested through the composition and activity of the ~1000 mitochondrial proteins, which are regulated at multiple points from transcription to degradation (*Bohovych et al., 2015*; *Fox, 2012*). In the area of mitochondrial protein degradation, cells are equipped with numerous proteolytic pathways that function to maintain mitochondrial homeostasis (*Anand et al., 2013*). Some of these pathways target individual mitochondrial proteins, whereas others act more broadly and remove large portions of the organelle. Examples of the former include AAA$^+$-ATPase proteases within all mitochondrial compartments that promote the degradation of unassembled protein complex subunits, oxidatively damaged proteins, and tail-anchored membrane proteins mistargeted to the mitochondrial outer membrane (*Chen et al., 2014*; *Gerdes et al., 2012*; *Okreglak and Walter, 2014*). There are also proteasome-dependent pathways

**eLife digest** Our cells contain compartments called mitochondria, which provide energy and serve as a home for many metabolic pathways that are critical for life. Changes in mitochondrial activity can contribute to aging and to the development of several age-associated diseases. However, our cells contain systems that detect changes in the performance of mitochondria and can act to re-establish a healthy state. These systems respond to stress or changes in metabolism by degrading particular proteins in the mitochondria. This enables damaged or unwanted proteins to be replaced with new proteins, but it is not clear how these mitochondrial defense systems work to keep mitochondria healthy as an organism ages.

Budding yeast is a good model in which to study the aging process because the performance of this yeast's mitochondria change in a characteristic way as the yeast ages. Previous studies have shown that these changes are caused by alterations in another cell structure called the lysosome, which can store nutrients and is where many proteins are degraded.

Here, Hughes et al. used yeast cells to investigate how mitochondrial defense systems operate during aging. The experiments reveal an entirely new mitochondrial protein degradation system that helps to keep the mitochondria healthy as the yeast cells age. This process is rapidly triggered by changes in lysosome activity and results in certain proteins in each mitochondrion being sorted into a small compartment made from a part of the mitochondrion. This compartment is released from the mitochondrion and then travels to the lysosome where the proteins are destroyed. Inhibiting the formation of these compartments results in mitochondria being more sensitive to cellular stress.

The next steps following on from this work are to find out exactly what role this mitochondrial defense pathway plays in cells and why it targets only a small set of all the proteins found in mitochondria.

that promote the turnover of proteins in the mitochondrial outer membrane and within internal mitochondrial compartments (*Taylor and Rutter, 2011*).

The best-studied pathway for destruction of large portions of mitochondria is through selective autophagy, or mitophagy (*Youle and Narendra, 2011*). In metazoa, the PINK1/Parkin pathway serves as a paradigm of mitophagy (*Pickrell and Youle, 2015*). PINK1 is a mitochondrial-localized kinase that is rapidly turned over in healthy mitochondria, but becomes stabilized on the surface of mitochondria with a reduced mitochondrial membrane potential, or $\triangle\Psi$ (*Narendra et al., 2010*). Upon stabilization, PINK1 recruits the E3 ligase Parkin, which ubiquitinates a number of mitochondrial surface proteins to promote fragmentation and destruction of dysfunctional mitochondria in the lysosome by autophagy (*Narendra et al., 2008*). Importantly, all the general machinery of autophagy must be active for mitophagy to occur. The apparent specificity comes in tagging mitochondria for autophagy.

The budding yeast, *S. cerevisiae*, contains no obvious sequence homologs of PINK1 and Parkin, but there are multiple reports of mitophagy occurring in yeast when cells are metabolically challenged or when $\triangle\Psi$ is compromised by genetic mutation or chemical treatment (*Kanki et al., 2011*). The most clearly defined version of mitophagy in budding yeast requires a mitochondrial outer membrane protein, Atg32, to link fragments of mitochondria to the autophagy machinery for degradation (*Kanki et al., 2009b*; *Okamoto et al., 2009*). Unlike the PINK1/Parkin pathway, the Atg32 pathway does not appear to respond to loss of $\triangle\Psi$ and does not utilize ubiquitin tagging for mitochondrial destruction. Instead, expression of Atg32 on the surface of mitochondria promotes turnover of mitochondria, primarily during times of regulated metabolic remodeling such as during the transition from robust growth to a starved or stationary phase-like state. In this regard, Atg32-dependent mitophagy may be a functional homologue to the tagging of mitochondria by the NIX protein during reticulocyte maturation (*Novak et al., 2010*). NIX is proposed to serve as a receptor on mitochondria to mediate mitophagy in this developmental process.

More recently, a second mode of eliminating large portions of mitochondria was discovered in mammalian cells. It involves the formation of mitochondrial-derived vesicles (MDVs), which deliver oxidized mitochondrial proteins to the lysosome for degradation (*Soubannier et al., 2012a*). PINK1/

Parkin are also needed for MDV formation and degradation, but interestingly the core autophagy machinery is not required (*McLelland et al., 2014*). This has led to the speculation that MDVs may provide an early wave of mitochondrial protein quality control, which if unsuccessful in re-achieving cellular homeostasis, is followed by 'full blown' mitophagy (*Sugiura et al., 2014*).

A key area in which many of these pathways function to promote mitochondrial homeostasis is during the aging process. There has long been a strong association between cellular aging and mitochondrial dysfunction (*Gonzalez-Freire et al., 2015*). However, there is an incomplete understanding of how these degradation systems contribute to promoting mitochondrial health in aging organisms. While mutations in PINK1/Parkin lead to early onset of neurodegeneration (*Kitada et al., 1998*; *Valente et al., 2004*), it is not yet clear what events trigger this dependence on the PINK1/Parkin pathway during the aging process. In fact, much of what is known about mitophagy and MDVs is based primarily upon acute stresses placed on mitochondria by chemical challenges or severe genetic perturbations (e.g. treatments that cause global ROS damage or complete loss of $\triangle\Psi$) (*Sugiura et al., 2014*).

Replicative aging of *S. cerevisiae*, defined by the number of times an individual yeast cell produces a daughter cell, functions as a model system for understanding fundamental aspects of cellular aging, including age-dependent mitochondrial dysfunction (*Breitenbach et al., 2014*; *Steinkraus et al., 2008*; *Wasko and Kaeberlein, 2014*). We and others have shown that aged yeast cells have reduced $\triangle\Psi$, decreased mitochondrial import, and altered mitochondrial morphology that transitions from a typical tubular shape to one that is highly fragmented and amorphous with increasing age (*Hughes and Gottschling, 2012*; *Lam et al., 2011*; *McFaline-Figueroa et al., 2011*; *Scheckhuber et al., 2007*). This change in mitochondrial structure and function is driven at least in part by changes in the pH of the yeast lysosome-like vacuole that occur earlier in the yeast cell's age (*Hughes and Gottschling, 2012*), as well as asymmetric partitioning of healthy mitochondria to daughter cells (*McFaline-Figueroa et al., 2011*). In the present study, we followed the fate of these age-associated changes in mitochondria to further our understanding of how cells maintain mitochondrial homeostasis during the course of aging. As described here, our studies led to the discovery of a new mitochondrial protein degradation pathway that selectively remodels the mitochondrial proteome to maintain optimum mitochondrial function.

## Results

### Mitochondrial proteins are degraded by autophagy in aged cells

As described above, autophagy systems play an important role in maintaining mitochondrial health across a variety of organisms. Therefore we tested whether autophagy played a role in maintaining mitochondrial function in aged yeast. Specifically, we examined aged cells for the presence of a mitochondrial outer membrane protein, Tom70-GFP, within vacuoles. No GFP signal was detected in the vacuole of wild-type cells across a wide range of ages (data not shown). However, vacuolar proteases rapidly degrade proteins taken up by autophagy and thus frequently prevent protein detection in the vacuole. Therefore, we re-assessed the presence of vacuolar Tom70-GFP in aged cells lacking *PEP4*, a gene encoding a master vacuolar protease that is required for turnover of autophagosomes in the vacuole (*Klionsky et al., 1992*; *Takeshige et al., 1992*). In young cells, which had functional mitochondria, no Tom70-GFP was detected in vacuoles by fluorescence microscopy (*Figure 1A*). However, vacuoles in 70% of aged cells contained Tom70-GFP (marked by white arrows in *Figure 1A*). Tom70-GFP in the vacuole appeared as a body exhibiting constant Brownian motion within the boundary of the vacuole membrane (data not shown), which is characteristic of autophagosomes that cannot be broken down in *pep4△* cells (*Takeshige et al., 1992*).

To test if the appearance of Tom70-GFP within the vacuole was autophagy-dependent, Tom70-GFP localization was examined in aged cells lacking *ATG5*, a gene essential for all forms of autophagy (*Feng et al., 2014*). Aged *atg5△ pep4△* cells contained no Tom70-GFP in the vacuole, which indicated that Tom70-GFP normally entered the vacuole via autophagy (*Figure 1A*). In addition to the core autophagy machinery, some forms of mitochondrial autophagy require the mitochondrial fission machinery (*Muller et al., 2015*). Delivery of Tom70-GFP to the vacuole in aged cells also required the mitochondrial fission machinery, as this process was inhibited in cells lacking *DNM1*, which encodes a conserved GTPase required for mitochondrial fission (*Figure 1A*) (*Bleazard et al.,*

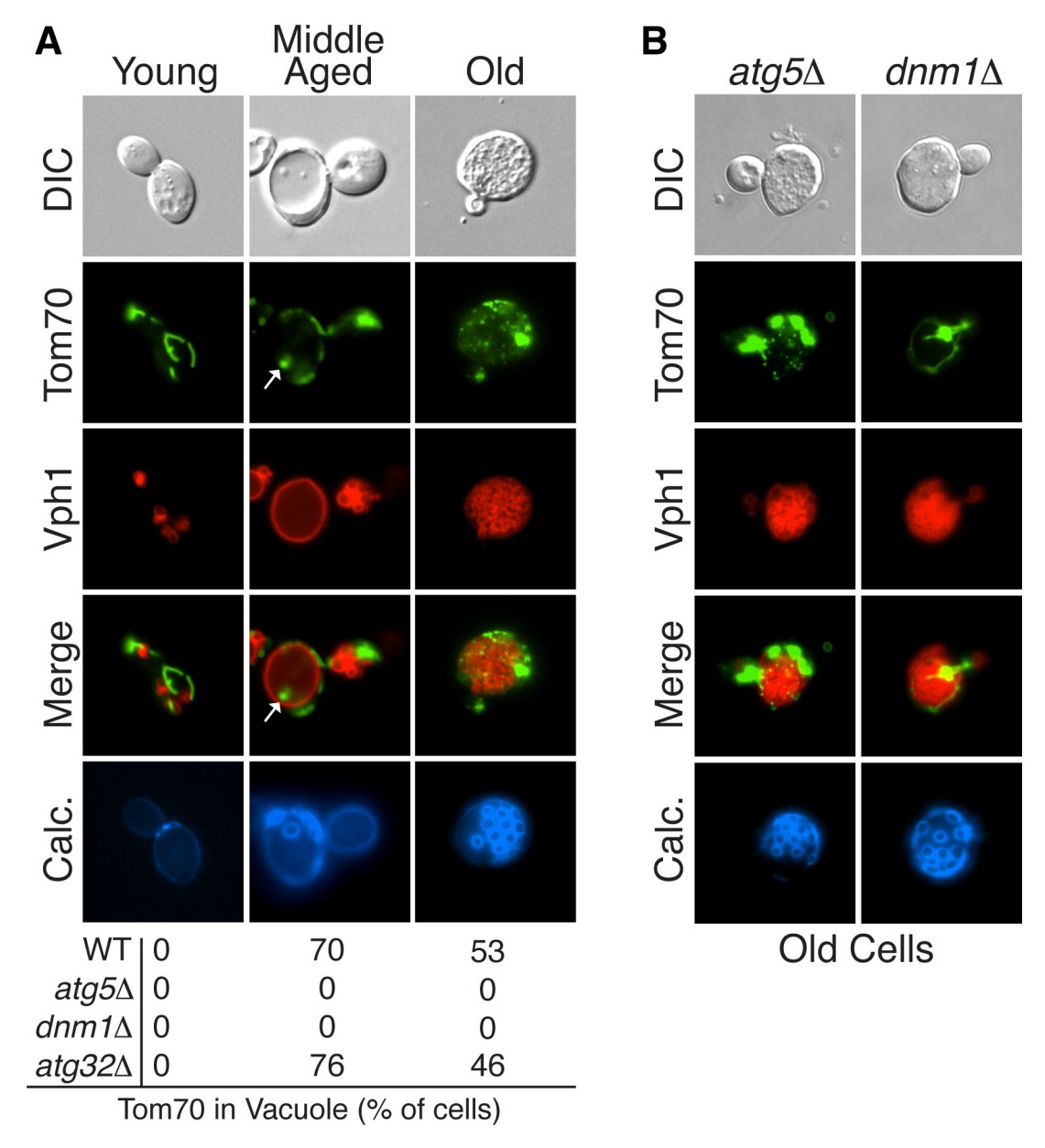

**Figure 1.** Mitochondrial proteins are degraded by autophagy in aged cells. (**A**) Tom70-GFP is degraded in the vacuole by autophagy in middle-aged cells. Wild-type (WT) and the indicated mutant cells expressing Tom70-GFP and the vacuole marker Vph1-mCherry were aged and visualized by fluorescence microscopy. Images depict wild-type cells, and the presence of Tom70-GFP in the vacuole (white arrow) of young, middle-aged, and old cells was scored for each strain. All strains including wild type are *PEP4*-deficient (*pep4△*). N = 30. In all figures, young cells have undergone 0–3 divisions, middle-aged cells 7–12 divisions, and old cells >17 divisions. Divisions are scored by counting bud scars visualized with calcolfuor (Calc). (**B**) Representative images of old *ATG5*- (*atg5△*) and *DNM1*-deficient (*dnm1△*) cells from (**A**) with fragmented vacuole morphology.

*1999*). Lastly, starvation-induced mitophagy in yeast relies on Atg32, a receptor on the mitochondrial surface that links mitochondria to the autophagy machinery for degradation (*Kanki et al., 2009b*; *Okamoto et al., 2009*). However, Tom70 degradation in aged cells is independent of Atg32; aged cells lacking *ATG32* delivered Tom70-GFP to the vacuole at the same level as in wild-type cells (*Figure 1A*). These results suggest we identified an autophagy-dependent pathway for degrading Tom70 in mitochondria from aging yeast cells that requires mitochondrial fission and is distinct from the previously characterized Atg32-dependent pathway.

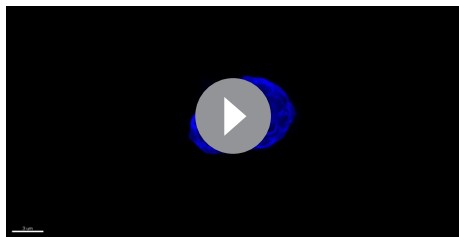

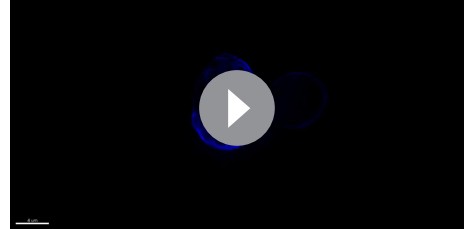

**Video 1.** 3D reconstruction of mitochondria and vacuoles in old wild-type cells. A representative 3D reconstruction of an old wild-type cell with the same characteristics as those depicted in *Figure 1A* showing small mitochondrial vesicle-like fragments (green, marked with Tom70-GFP) outside of the severely fragmented vacuole (red, marked with Vph1-mCherry). To permit visualization of the vacuole lumen, the vacuole isosurface rendering becomes 60% transparent in the middle of the movie. Budscars (blue, calcofluor) at the beginning of the movie indicate the cell's old age.

**Video 2.** 3D reconstruction of mitochondria and vacuoles in old *atg5△* cells. A representative 3D reconstruction of an old *atg5△* cell with the same characteristics as those depicted in *Figure 1B* showing small mitochondrial vesicle-like fragments (green, marked with Tom70-GFP) outside of the severely fragmented vacuole (red, marked with Vph1-mCherry). To permit visualization of the vacuole lumen, the vacuole isosurface rendering becomes 60% transparent in the middle of the movie. Budscars (blue, calcofluor) at the beginning of the movie indicate the cell's old age.

While examining the kinetics of Tom70-GFP vacuolar delivery in aged cells, we noticed that the presence of Tom70 in the vacuole peaked in middle-aged cells, and then declined in old cells (*Figure 1A*). This decline coincided with a highly fragmented vacuole, depicted in the third panel of *Figure 1A*. For reasons that are unclear, vacuoles increase in size with age and then can become severely fragmented in very old cells (*Figure 1A*) (*Lee et al., 2012*). Interestingly, in all cells with a severely fragmented vacuole, Tom70-GFP appears in small vesicle-like structures in the cytoplasm (*Figure 1A* and *Video 1*). These structures are not cytosolically-localized autophagosomes, because they are still present in old cells lacking *ATG5* (*Figure 1B* and *Video 2*). However, their formation does require the mitochondrial fission GTPase *DNM1* (*Figure 1B* and *Video 3*). Thus, although mitochondrial protein destruction is activated in middle-aged cells, this autophagy-dependent degradation appears compromised in very old yeast, leading to production of Dnm1-dependent vesicle-like structures.

## Loss of vacuolar acidity triggers mitochondrial protein degradation

We previously showed that mitochondrial dysfunction in aged cells is caused by disruption of a metabolic relationship between mitochondria and vacuoles (*Hughes and Gottschling, 2012*). Vacuoles are acidified by the Vacuolar-H$^+$-ATPase (V-ATPase) (*Kane, 2006*), and the proton gradient generated by this protein complex is required for amino acid storage within the vacuole lumen (*Klionsky et al., 1990*). Loss of vacuole acidity in aged cells causes mitochondrial dysfunction through an undefined mechanism that likely involves altered storage of cellular amino acids (*Hughes and Gottschling, 2012*). To test if loss of vacuole acidity triggers autophagy-dependent mitochondrial protein degradation, we took advantage of the fact that treatment of young cells with concanamycin A (conc A), a specific inhibitor of the V-ATPase (*Drose et al., 1993*), recapitulates age-associated changes in

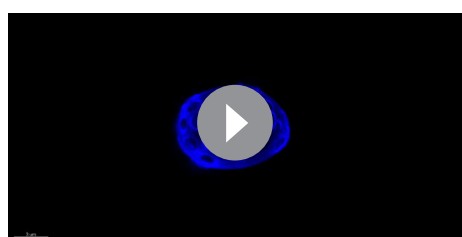

**Video 3.** 3D reconstruction of mitochondria and vacuoles in old *dnm1△* cells. A representative 3D reconstruction of an old *dnm1△* cell with the same characteristics as those depicted in *Figure 1B* showing the presence of mitochondria (green, marked with Tom70-GFP), but the absence of small mitochondrial vesicle-like fragments outside of the severely fragmented vacuole (red, marked with Vph1-mCherry). To permit visualization of the vacuole lumen, the vacuole isosurface rendering becomes 60% transparent in the middle of the movie. Budscars (blue, calcofluor) at the beginning of the movie indicate the cell's old age.

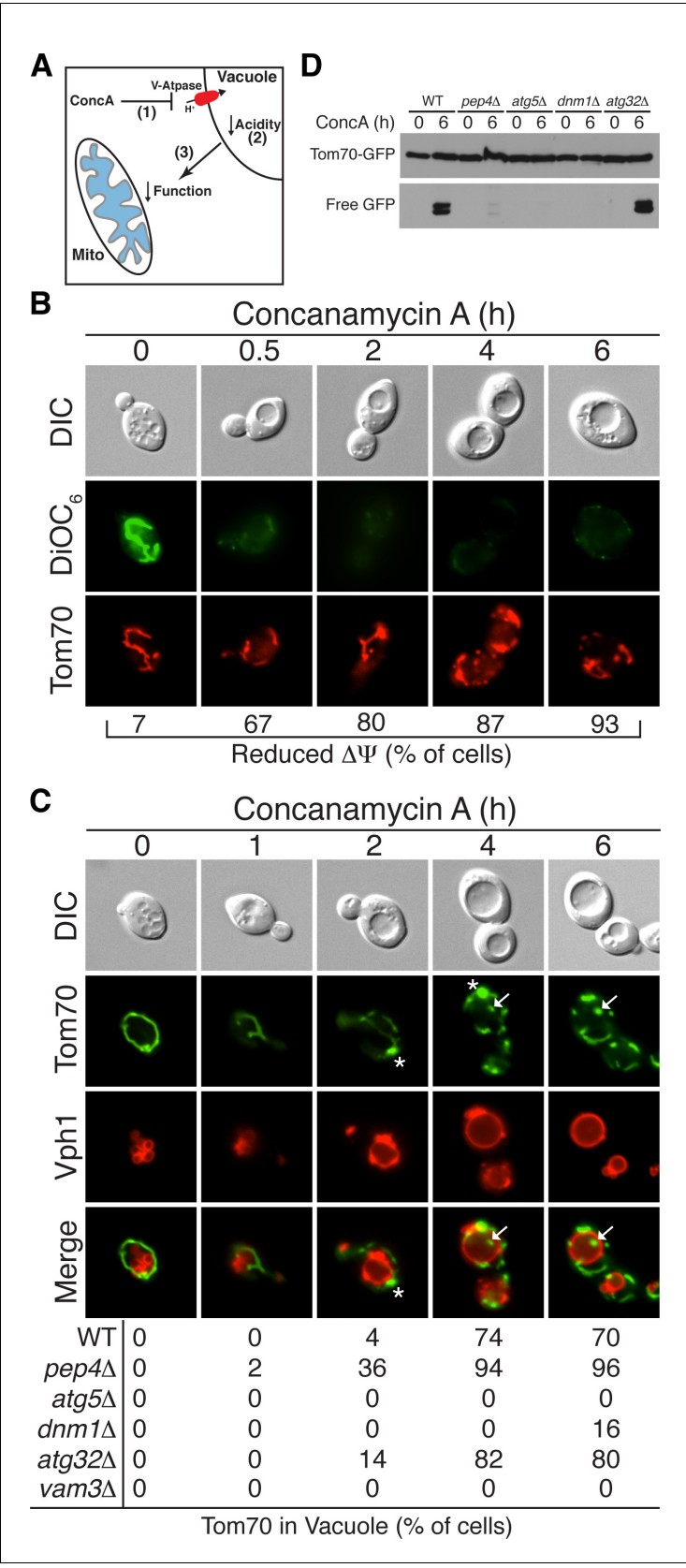

**Figure 2.** Loss of vacuole function triggers mitochondrial protein degradation. (**A**) Schematic illustration showing that loss of vacuolar acidity (2) through aging or concanamycin A (conc A)-mediated inhibition of the Vacuolar H$^+$-

*Figure 2 continued on next page*

*Figure 2 continued*

ATPase (1) leads to loss of mitochondrial function (3) through an unknown mechanism. (**B**) Loss of vacuolar acidity causes rapid mitochondrial depolarization. Wild-type cells expressing Tom70-mCherry were treated with concanamycin A for the indicated time (hr) and stained with $DiOC_6$ as an indicator of $\triangle\Psi$. N = 30. (**C**) Loss of vacuolar acidity activates autophagy-dependent Tom70-GFP degradation. Wild-type (WT) and the indicated mutant cells expressing Tom70-GFP and Vph1-mCherry were treated with concanamycin A for the indicated time (hr). The presence of Tom70-GFP in the vacuole (white arrow) was scored for each strain and time point. N = 50. * indicates MDC. (**D**) Tom70-GFP was monitored for autophagy-dependent degradation using a GFP-cleavage assay in wild-type (WT) and the indicated mutant cells treated with concanamycin A (ConcA) for the indicated time (hr). Whole-cell extracts from the treated cells were subjected to immunoblot analysis with anti-GFP antibody. The use of conc A as an inducer potentially limited the amount of GFP cleavage in the vacuole. Consequently, the exposure time of the free GFP immunoblot is 20 times longer than the exposure of the immunoblot with full length Tom70-GFP.

The following figure supplement is available for figure 2:

**Figure supplement 1.** Concanamycin a treatment causes loss of mitochondrial membrane potential.

mitochondria (*Figure 2A*) (*Hughes and Gottschling, 2012*). Consistent with our previous findings, treatment with conc A caused an immediate loss of vacuolar acidity, followed by a decline in mitochondrial $\triangle\Psi$ within 30 min as measured by microscopy (*Figure 2B*) using the common mitochondrial membrane potential fluorescent dye $DiOC_6$ (*Pringle et al., 1989*), or by flow cytometry (*Figure 2—figure supplement 1*) using $DiOC_6$ and another mitochondrial membrane potential dye, TMRM (*Scaduto and Grotyohann, 1999*). Similar to our observations in aged cells, treatment of young cells with conc A caused accumulation of Tom70-GFP-containing bodies within the vacuole (*Figure 2C*, white arrows). Because no Tom70-GFP was present in the vacuoles of untreated *pep4△* cells, we conclude that loss of vacuolar acidity triggers activation of this pathway. Additionally, in contrast to old cells, it was not necessary to delete *PEP4* to observe Tom70-GFP within the vacuole of conc A treated cells. This is likely because inhibition of the V-ATPase with conc A creates an environment within the lumen that prevents autophagosome breakdown (*Nakamura et al., 1997*). The appearance of Tom70-GFP in the conc A treated vacuole occurred within three hours after mitochondrial depolarization, and was preceded by the formation of a single bright focus that contained Tom70-GFP, but appeared somewhat distinct from the rest of the mitochondria (*Figure 2C*, white *). As in old cells, delivery of Tom70-GFP to the vacuole was dependent on both the autophagy (Atg5) and mitochondrial fission machineries (Dnm1), but independent of Atg32 (*Figure 2C*). Further supporting the role of autophagy in this process, delivery of Tom70-GFP to the vacuole also required the vacuole homotypic fusion protein Vam3, which is essential for fusion of autophagosomes with the vacuole (*Figure 2C*) (*Darsow et al., 1997*).

In addition to the microscopy-based assays, turnover of GFP-tagged proteins by autophagy can also be detected by monitoring release of free GFP from the epitope-tagged protein in the vacuole using immunoblot analysis (*Kanki et al., 2009a*). Once released by vacuolar proteases, GFP is stable in the vacuole lumen. The release of GFP from Tom70-GFP was monitored by immunoblot analysis in cells treated with conc A, and the results mirrored what was observed in the microscopy-based assay (*Figure 2D*). Treatment of cells for six hours caused release of GFP from full length Tom70-GFP (*Figure 2D*). GFP cleavage required the vacuolar protease *PEP4*, the autophagy and mitochondrial fission machineries, and was independent of *ATG32* (*Figure 2D*). Collectively, these results suggest that disruption of the mitochondrial-vacuole relationship by direct V-ATPase inhibition activates autophagy-dependent mitochondrial protein degradation.

## Tom70-GFP degradation is not triggered by loss of mitochondrial membrane potential or oxidative stress

The timing of Tom70-GFP destruction shortly after mitochondrial depolarization raised the possibility that loss of mitochondrial membrane potential may activate this pathway downstream of changes in vacuole acidity. To test this, we analyzed Tom70-GFP vacuolar delivery in cells treated with two common mitochondrial depolarizing agents, the electron transport chain inhibitor antimycin A, and the proton ionophore carbonyl cyanide-4-(trifluoromethoxy)phenylhydrazone (FCCP). Although

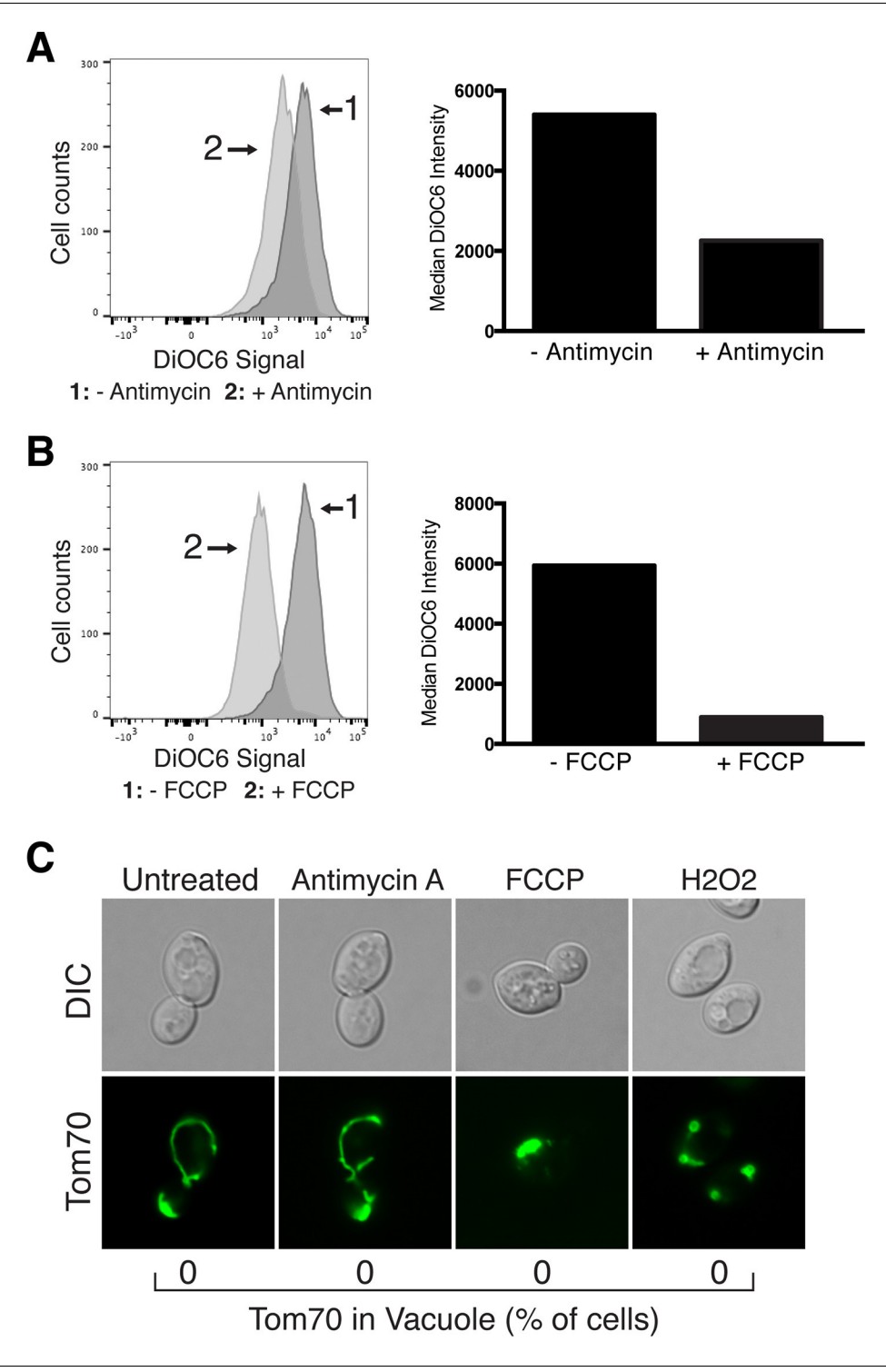

**Figure 3.** Mitochondrial protein degradation is not triggered by loss of mitochondrial membrane potential or oxidative stress. Antimycin A and FCCP cause mitochondrial depolarization. Wild-type cells were treated with Antimycin A (**A**) or FCCP (**B**) for 4 hr, stained with the mitochondrial membrane potential fluorescent dye DiOC$_6$, and analyzed by flow cytometry. FACS profiles and bar graphs showing median fluorescence intensity are shown for each treatment. N > 10,000 cells for each. (**C**) Loss of mitochondrial membrane potential or oxidative stress does not activate autophagy-dependent Tom70-GFP degradation. Wild-type cells expressing Tom70-GFP were treated with the indicated compound for 4 hr. The presence of Tom70-GFP in the vacuole was scored for each strain and time point. N = 50. Representative images showing mitochondrial aggregation and fragmentation in FCCP and hydrogen peroxide (H$_2$O$_2$) treated cells are shown.

treatment of cells with these inhibitors caused mitochondrial depolarization equal to or greater than conc A (*Figure 3A and B*), we did not observe Tom70-GFP delivery to the vacuole at any point during treatment with these compounds (*Figure 3C*). Instead, mitochondria coalesced in FCCP treated cells, and were largely unchanged with antimycin A treatment (*Figure 3C*). These results suggest that a decrease in mitochondrial membrane potential is not sufficient to trigger Tom70-GFP degradation.

In addition to mitochondrial depolarization, loss of vacuolar acidity also causes oxidative stress in cells through an undefined mechanism (*Milgrom et al., 2007*). Therefore, we also tested whether oxidants trigger activation of this pathway. Treatment of cells with hydrogen peroxide caused mitochondrial fragmentation, but like FCCP and antimycin A, did not lead to the appearance of Tom70-GFP within the vacuole (*Figure 3C*). This result suggests that oxidative stress is also not sufficient to activate this pathway, and that loss of vacuolar acidity triggers mitochondrial protein degradation through an unknown mechanism.

## The mitochondrial-derived compartment (MDC) is an intermediate step in mitochondrial protein degradation

During the analysis of mitochondrial protein degradation in young cells, we noticed a distinct structure containing Tom70-GFP that formed prior to the appearance of Tom70-GFP foci within the vacuole (*Figure 2C*, white *). We refer to this structure as the mitochondrial-derived compartment (MDC). MDCs appeared attached to mitochondria as a single focus with greater signal intensity than the rest of the organelle. These structures were also present in middle-aged cells undergoing mitochondrial degradation (*Figure 4A*), and the timing of their appearance suggested that MDCs are an intermediate in this process.

To further analyze the kinetics of MDC formation during the degradation process, MDCs were monitored during a timecourse of conc A treatment. In wild-type cells, MDCs appeared shortly after mitochondrial depolarization, and were present in a large percentage of cells within two hours after conc A addition (*Figure 4B*). Cells generally contained only a single MDC, and it often localized near the vacuolar membrane prior to the appearance of Tom70-GFP inside the vacuole lumen (*Figure 4B*, white arrows). MDCs formed independently of both autophagy and mitochondrial fission (*Figure 4B*). However, although fission-defective *dnm1Δ* cells formed MDCs normally at two hours after conc A addition, at later timepoints, MDCs grew very large in these cells and were not released from mitochondria to the vacuole (*Figure 4C*, white arrows). This suggests that mitochondrial membrane fission activity is required for release of MDCs for autophagic degradation. In support of this conclusion, Fis1, an essential component of the mitochondrial fission machinery that recruits Dnm1 to mitochondrial membrane (*Mozdy et al., 2000*), is also required for release of MDCs from mitochondria (*Figure 4—figure supplement 1A*). Moreover, foci of Dnm1-GFP, representing sites of mitochondrial fission, localize near MDCs in 88% of cells (*Figure 4—figure supplement 1B*). Collectively, these data suggest that the mitochondrial fission machinery is required for MDC release.

Like the fission mutants, autophagy-deficient mutants (*atg5△*) formed normal MDCs after two hours of conc A treatment (*Figure 4C*, white arrows), suggesting that autophagy is not required for MDC formation. However, at later timepoints, Tom70-GFP did not enter the vacuole in these mutants, and MDCs were less apparent. Instead, Tom70-GFP was present in small vesicle-like structures (*Figure 4C*) that looked similar to those that formed in very old cells (*Figure 1A*). Further supporting the role of autophagy in MDC degradation, MDCs in 68% of cells localized at or near the preautophagosomal structure (PAS) at the vacuole membrane, which is the site of autophagosome formation and target engulfment in yeast (*Figure 4—figure supplement 2*) (*Feng et al., 2014*). Taken together, these results suggest that the MDC is an intermediate step on the way to Tom70-GFP degradation in the vacuole by autophagy, and that its release from the mitochondrial surface relies on mitochondrial fission. At this point, we also cannot rule out other alternative fates for the MDC, such as delivery to other cellular compartments.

## Protein incorporation into MDCs is selective

During characterization of the MDC pathway in young cells, we found that unlike Tom70, the inner mitochondrial membrane protein Tim50-GFP was excluded from MDCs and not targeted for

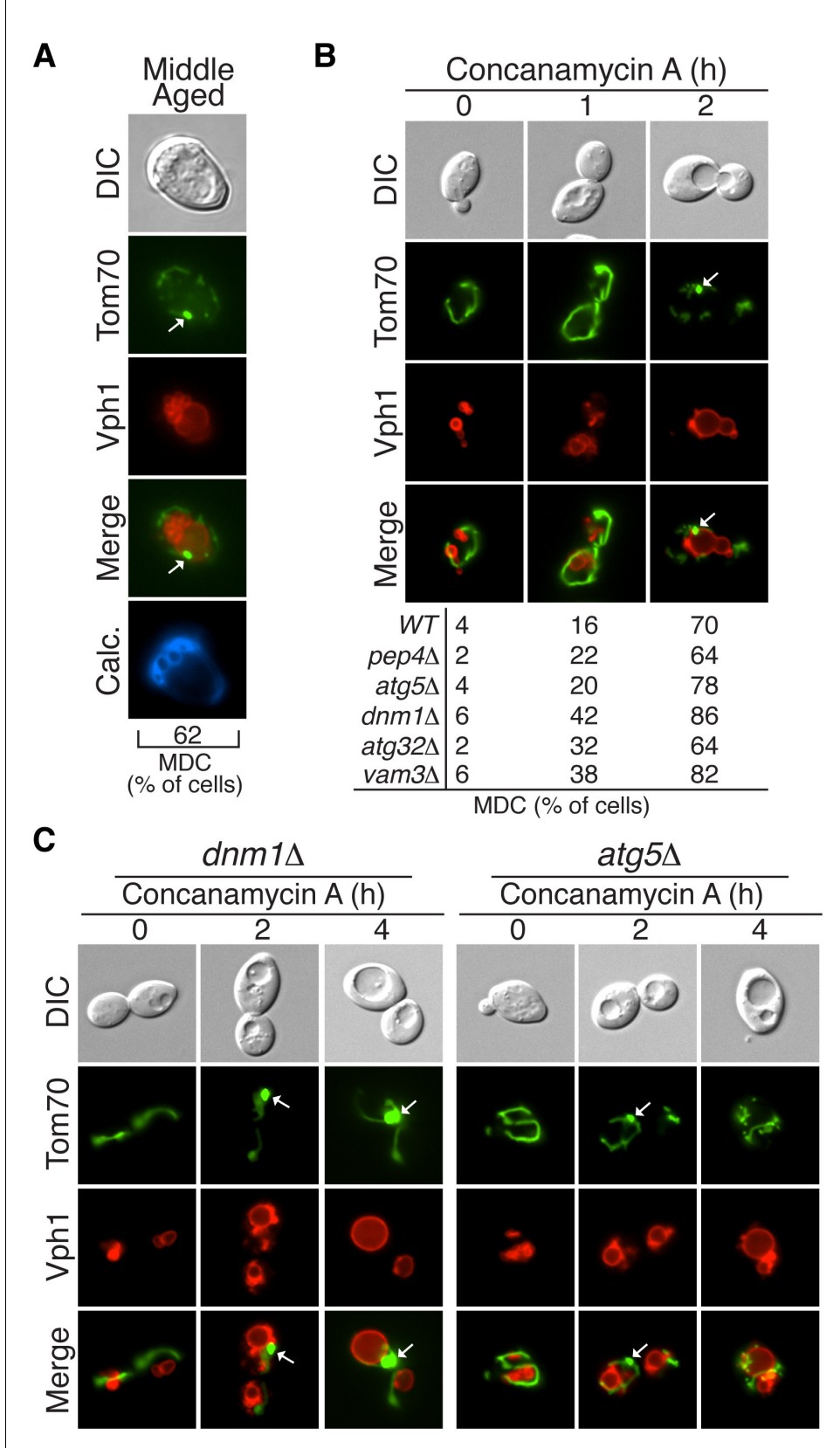

**Figure 4.** The mitochondrial-Derived compartment (MDC) is an intermediate step in mitochondrial protein degradation. (**A**) Aging induces MDC formation. Middle-aged cells expressing Tom70-GFP and Vph1-mCherry

*Figure 4 continued on next page*

*Figure 4 continued*

were scored by fluorescence microscopy for the presence of mitochondrial-derived compartment (MDC) structures (white arrow). N = 30. (B) Loss of vacuolar acidity triggers MDC formation. Wild-type (WT) and the indicated mutant cells expressing Tom70-GFP and Vph1-mCherry were treated with concanamycin A for the indicated time (hr). The presence of MDCs (white arrow) was scored for each strain and time point. N = 50. (C) Mitochondrial fission, but not autophagy, is required for MDC release. Representative images of MDCs (white arrow) in *DNM1*-(*dnm1△*) and *ATG5*-deficient (*atg5△*) cells from (B).

The following figure supplements are available for figure 4:

**Figure supplement 1.** Further support for the role of fission proteins in MDC release from the mitochondria.

**Figure supplement 2.** Further support for the role of autophagy in Tom70-GFP degradation.

vacuolar degradation by this pathway (*Figure 5A and B*). Similar specificity was apparent in aged cells (*Figure 5C*), suggesting that MDCs are cargo-selective.

To understand the extent of cargo-selectivity, a microscopy-based screen was performed to identify the proteins that are incorporated into MDCs. For the screen, we created a collection of yeast strains coexpressing Tom70-mCherry and any protein of interest fused to GFP by crossing a strain containing Tom70-mCherry to the yeast GFP strain collection (*Huh et al., 2003*). Strains containing GFP-tagged mitochondrial-localized proteins were screened after conc A treatment in a similar manner to the Tim50-GFP/Tom70-mCherry strain in *Figure 5A*. In total, 469 mitochondrial proteins were examined (*Supplementary file 1*) and 304 of the proteins were detectable by microscopy. Of those, 26 localized to MDCs and were degraded in the vacuole (*Table 1*), indicating that this pathway exhibits narrow substrate specificity. Although the MDC incorporated a relatively small number of the total proteins examined, it is likely specific for mitochondrial proteins, because markers of other major organelles were not co-localized with MDCs (*Figure 5—figure supplement 1*).

Mitochondrial proteins broadly localize to four distinct mitochondrial subdomains: the outer membrane, inner membrane, inner membrane space, and matrix (*Fox, 2012*). There was subdomain specificity in mitochondrial proteins that were targeted to the MDC. Representative examples of substrates and non-substrates from various mitochondrial subcompartments are shown in *Figure 5D*. Nearly all integral- and peripherally-associated mitochondrial outer membrane proteins, as well as a subclass of mitochondrial inner membrane proteins belonging to the mitochondrial carrier family localized to the MDC. In contrast, mitochondrial matrix proteins and inner membrane proteins that were not part of the carrier family were excluded. Because they were below the detection limit of the assay, it was not possible to ascertain whether mitochondrial inner membrane space proteins or β-barrel proteins of the outer membrane were present in MDCs.

The MDC substrate specificity followed classic lines of mitochondrial import (*Schmidt et al., 2010*). Many matrix and inner membrane-localized mitochondrial proteins contain a mitochondrial targeting sequence that is removed upon import into the mitochondria. After being translated in the cytoplasm, this class of presequence-containing proteins generally binds to the mitochondrial surface receptor Tom20, and then are routed through the TOM complex of the outer membrane and the Tim23 complex of the inner membrane to their final destination. Proteins that are normally imported via this pathway were excluded from MDCs (*Figure 5D* and *Supplementary file 1*).

In contrast to presequence-containing proteins, integral inner membrane carrier proteins of the metabolite carrier family and alpha-helical outer membrane proteins lack a cleavable mitochondrial presequence (*Harbauer et al., 2014*). After translation, these proteins bind to the Tom70 mitochondrial surface receptor, and then are targeted to the inner membrane through the Tim22 complex or to the outer membrane through the Mim1 pathway (*Becker et al., 2011*). All detectable proteins that rely on Tom70-associated import pathways were targeted to MDCs, including Tom70 itself (*Table 1*). In further support of the distinction between the Tom20 and Tom70 import pathways, Tom20 was excluded from MDCs (*Figure 5E*), even though it is an integral outer membrane protein (*Schneider et al., 1991*). Even Cox7, which is not a carrier protein, but lacks a cleavable presequence and thus likely utilizes Tom70 for mitochondrial import was targeted to MDCs (*Table 1*) (*Calder and McEwen, 1990*; *Mihara and Blobel, 1980*). Thus, the MDC pathway only degraded a

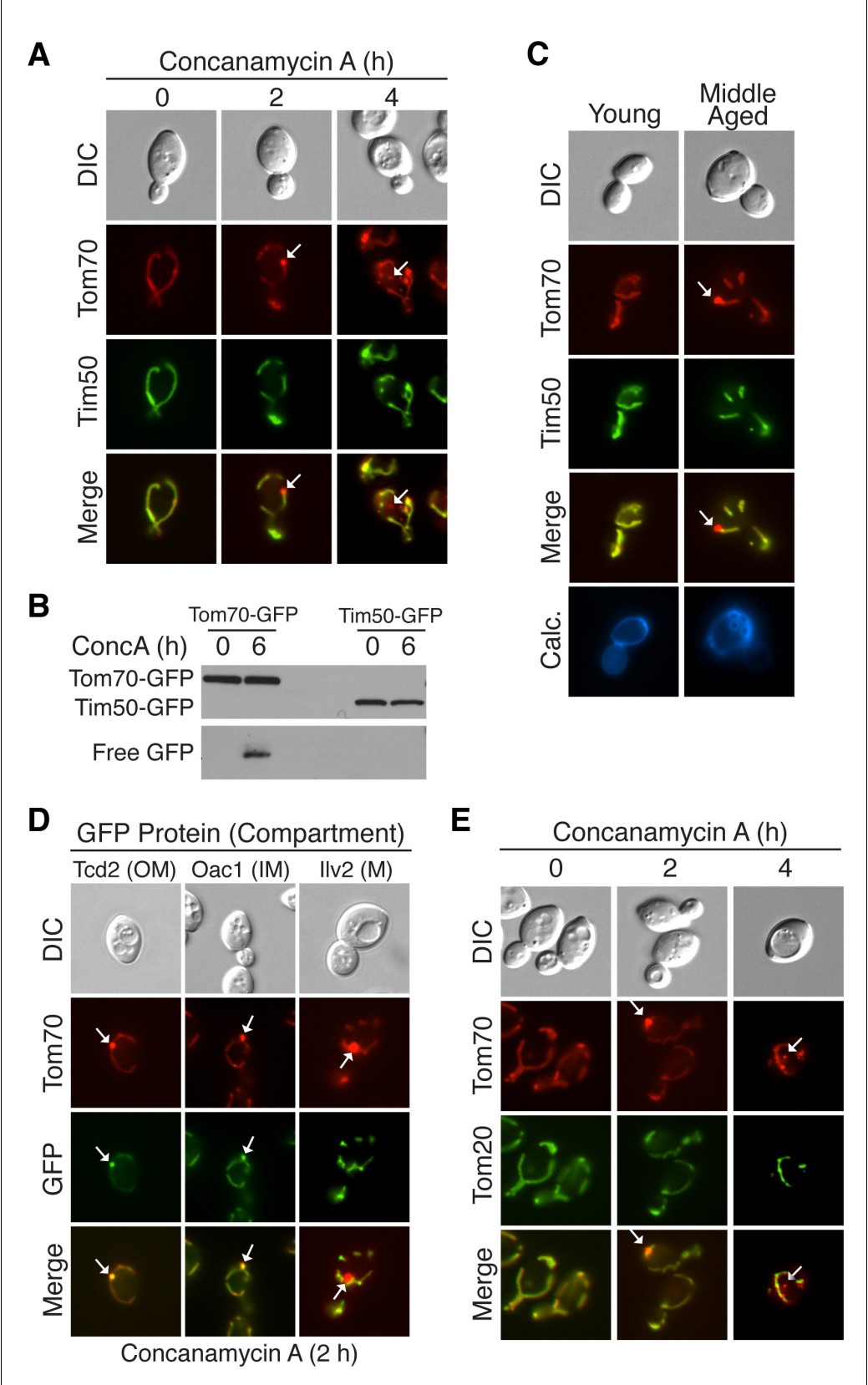

**Figure 5.** Select mitochondrial proteins are incorporated into MDCs. (**A**) The inner mitochondrial membrane protein Tim50 is excluded from MDC-dependent degradation. Wild-type cells expressing Tom70-mCherry and Tim50-GFP were treated with concanamycin A for the indicated time and
*Figure 5 continued on next page*

*Figure 5 continued*

representative images showing Tim50 exclusion from the MDC (2 hr, white arrow) and vacuole (4 hr, white arrow) are shown. 100% of cells show this phenotype of Tim50 exclusion. N = 50. (B) Tom70-GFP and Tim50-GFP were monitored for autophagy-dependent degradation using a GFP-cleavage assay in wild-type (WT) cells expressing either Tom70-GFP or Tim50-GFP treated with concanamycin A (ConcA) for the indicated time (hr). Whole-cell extracts from the treated cells were subjected to immunoblot analysis with anti-GFP antibody. As in *Figure 2D*, the use of conc A as an inducer limited the amount of GFP cleavage in the vacuole. Consequently, the exposure time of the free GFP immunoblot is ~20 times longer than the exposure of the immunoblot with full-length GFP-tagged proteins. (C) Tim50 is excluded from MDCs in middle-aged cells. Wild-type cells expressing Tom70-mCherry and Tim50-GFP were aged and representative images showing Tim50 exclusion from the MDC (white arrow) are shown. 100% of cells show this phenotype of Tim50 exclusion. N = 30. (D) Mitochondrial outer membrane proteins and inner membrane carrier proteins localize to MDCs. Wild-type cells expressing Tom70-mCherry and the indicated C-terminal GFP fusion proteins were treated with concanamycin A for 2 hr and protein inclusion in MDCs (white arrows) was assessed. GFP-tagged marker proteins represent mitochondrial outer membrane (OM), inner membrane carrier proteins (IM), and matrix proteins (M). 100% of cells show the phenotypes in the representative images. N = 50. (E) Tom20 is excluded from MDCs. Wild-type cells expressing Tom70-mCherry and Tom20-GFP were treated with concanamycin A for the indicated time and representative images showing Tom20 exclusion from the MDC (2 hr, white arrow) and vacuole (4 hr, white arrow) are shown. 100% of cells show this phenotype of Tom20 exclusion. N = 50.

The following figure supplement is available for figure 5:

**Figure supplement 1.** MDCs do not contain other major organelles.

subset of mitochondrial proteins, and was specific for non-presequence containing proteins that rely on the Tom70 surface receptor for import.

## The MDC can target preexisting mitochondrial proteins

Because MDC substrates were specifically confined to Tom70-dependent import pathways, we wondered if the MDC was an aggregate of newly synthesized proteins that could not be imported into dysfunctional mitochondria. If this were true, proteins incorporated into MDCs would be newly synthesized, not preexisting mitochondrial proteins. To test this, old and newly synthesized versions of proteins were monitored for their incorporation into the same MDC using the recombination induced tag exchange (RITE) system (*Verzijlbergen et al., 2010*). As illustrated in *Figure 6A*, fusion of the RITE cassette to a protein of interest allows rapid and permanent switching of epitope tags through an estradiol-inducible Cre/loxP based mechanism. With this system, old protein synthesized before the switch is GFP tagged, while all subsequently synthesized protein is mRFP tagged. The RITE cassette was fused to Tom70 and Oac1, outer and inner membrane MDC substrates, respectively. Cells expressing these proteins were treated with conc A to induced MDC formation, along with estradiol to switch tags on the proteins from GFP to mRFP. Both mRFP and GFP versions of each protein were found in the MDC (*Figure 6B*), suggesting that MDCs incorporate old, preexisting proteins, and are not specific to unimported mitochondrial substrates. Because the mRFP-tagged protein targeted to MDCs may be synthesized prior to their formation, we cannot determine with this assay whether newly synthesized mitochondrial proteins are also targeted to this compartment. However, further supporting the idea that MDCs incorporate proteins that preexist within mitochondria, MDCs still formed in the presence of cycloheximide, which effectively prohibited synthesis of the newly synthesized version of both RITE-tagged Tom70 and Oac1 (*Figure 6C*). These results suggest that preexisting mitochondrial proteins are segregated into MDCs prior to degradation by autophagy.

## The mitochondrial import receptors Tom70 and Tom71 are required for MDC formation

Because the proteins degraded by the MDC pathway all rely on the mitochondrial import receptor Tom70 for import into mitochondria, we wondered whether Tom70 is required for MDC formation beyond its role in mitochondrial import (*Sollner et al., 1990*). To test this, we monitored the appearance of the MDC substrate Cox7-GFP (*Table 1*) in the vacuole of conc A treated cells. Cox7-GFP acts exactly the same as Tom70-GFP upon concanamycin A treatment, entering MDCs at 2 hr and getting delivered to the vacuole after four hours of treatment (*Figure 7—figure supplement 1A–B*). Despite the fact that Tom70 [and its paralog Tom71 (*Schlossmann et al., 1996*)] are likely required for import of Cox7 into the mitochondria (*Schlossmann et al., 1996*; *Schmidt et al., 2010*), a large

**Table 1.** MDC Substrates.

| Gene | ORF | Mitochondrial Localization[a] |
|---|---|---|
| NCA2 | YPR155C | Outer membrane |
| ALO1 | YML086C | Outer membrane |
| UBP16 | YPL072W | Outer membrane |
| MFB1 | YDR219C | Outer membrane |
| TCD2 | YKL027W | Outer membrane |
| TCD1 | YHR003C | Outer membrane |
| MCY1 | YGR012w | Outer membrane |
| MSP1 | YGR028W | Outer membrane |
| TOM70 | YNL121C | Outer membrane |
| TOM71 | YHR117W | Outer membrane |
|  | YPR098C | Outer membrane |
| OM14 | YBR230C | Outer membrane |
| SEN15 | YMR059W | Outer membrane |
| PTH2 | YBL057C | Outer membrane |
| MCP1 | YOR228C | Outer membrane |
| SCM4 | YGR049W | Outer membrane |
| MIR1[b] | YJR077C | Inner membrane |
| CTP1[b] | YBR291C | Inner membrane |
| DIC1[b] | YLR348C | Inner membrane |
| OAC1[b] | YKL120W | Inner membrane |
| MTM1[b] | YGR257C | Inner membrane |
| YMC2[b] | YBR104W | Inner membrane |
| YHM2[b] | YMR241W | Inner membrane |
| COX7 | YMR256C | Inner membrane |
|  | YML007C-A | Unknown |
| ECM19 | YLR390W | Unknown |

a, Localization information obtained from SGD (www.yeastgenome.org)
b, Mitochondrial carrier protein family member

amount of Cox7 still localized to mitochondria in both *tom70△* and *tom70△ tom71△* cells (*Figure 7A*). This import likely results from compensation by import receptors Tom20 and Tom22 (*Lithgow et al., 1994*; *Ramage et al., 1993*; *Steger et al., 1990*). Nevertheless, deletion of *TOM70* alone severely impaired conc A induced vacuolar delivery of Cox7 (*Figure 7B*), and loss of both Tom70 and 71 provided complete inhibition (*Figure 7B*).

To determine at which step in MDC-mediated degradation Tom70 and Tom71 function, we quantified the amount of Cox7-GFP present in MDCs from a conc A treated fission deficient strain (*dnm1△*). Loss of *TOM70* alone severely blocked MDC formation, and the loss of both *TOM70* and *TOM71* was completely inhibitory (*Figure 7C*). Collectively, these results suggest that in addition to their function as mitochondrial import receptors, Tom70 and Tom71 are also required for formation of MDCs and the destruction of mitochondrial proteins residing within them.

## Failure to form MDCs exacerbates loss of mitochondrial membrane potential

Because MDC formation coincided with depletion of the mitochondrial membrane potential (*Hughes and Gottschling, 2012*), we wondered whether MDC formation could protect mitochondria against further membrane potential loss. To test this, we monitored mitochondrial membrane

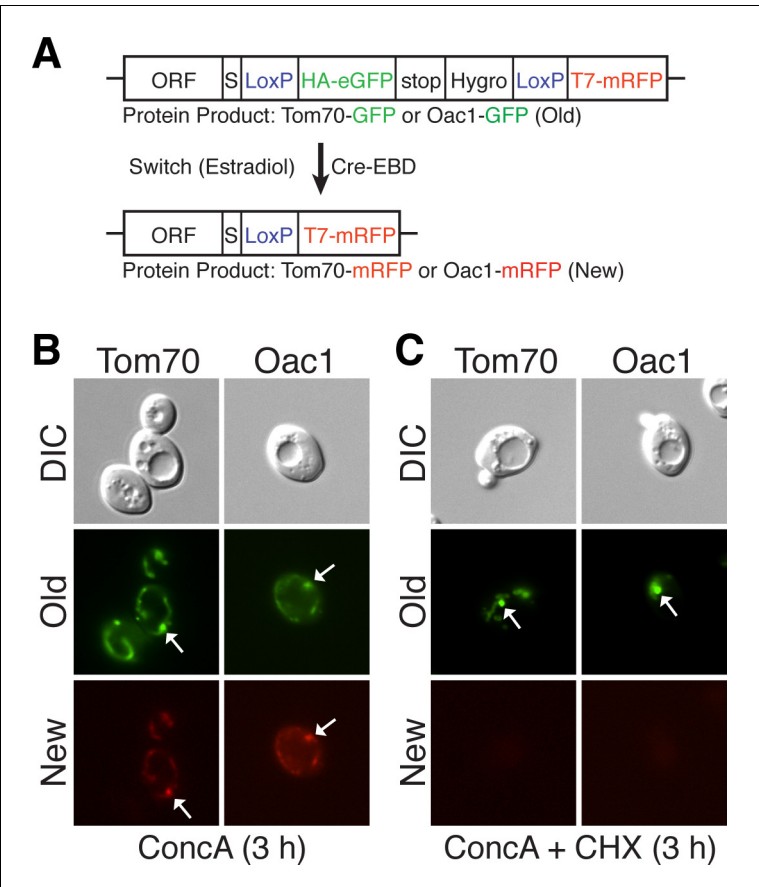

**Figure 6.** MDCs sequester preexisting mitochondrial proteins. (**A**) Schematic of the recombination induced tag exchange (RITE) system. In untreated cells, the RITE-tagged ORF is expressed with a C-terminal HA-GFP fusion (old protein). Treatment with estradiol induces Cre-EBD dependent recombination between LoxP sites creating a new C-terminal T7-mRFP fusion (new protein). (**B**) Preexisting protein is incorporated into MDCs. Cells expressing either RITE-tagged Tom70 or Oac1 were treated with estradiol and concanamycin A simultaneously for 3 hr to induce epitope tag exchange and MDC formation. Cells were visualized with fluorescence microscopy for the presence of preexisting (old) or newly synthesized (new) protein in the MDC (white arrows). 100% of the cells show the represented phenotype. N= 50. (**C**) MDCs (white arrows) form in the absence of new protein synthesis. Cells expressing RITE-tagged Tom70 or Oac1 were treated as in (**B**) with the addition of cycloheximide to inhibit synthesis of new T7-RFP tagged proteins. 100% of the cells show the represented phenotype. N= 50.

potential by flow cytometry in conc A treated cells using TMRM. As expected, wild-type cells showed a reduction in TMRM staining after conc A treatment (*Figure 7D* and *Figure 7—figure supplement 1C–D*). In the absence of conc A, cells lacking *TOM70* alone or both *TOM70* and *TOM71* had a similar membrane potential as wild-type cells. However, upon treatment with conc A, these cells showed a greater reduction in TMRM staining (*Figure 7D* and *Figure 7—figure supplement 1C–D*). At this time, we cannot rule out the possibility that further depletion of membrane potential in these strains results from a function of Tom70/71 unrelated to MDC formation. However, because these mutant strains cannot form MDCs under these conditions, it is possible that the sequestration of mitochondrial membrane proteins by MDCs may protect against mitochondrial depolarization caused by changes in vacuolar function.

## Discussion

Using yeast as a model, we have identified a new form of autophagy-dependent mitochondrial protein degradation that is activated in aged cells undergoing vacuole-induced mitochondrial

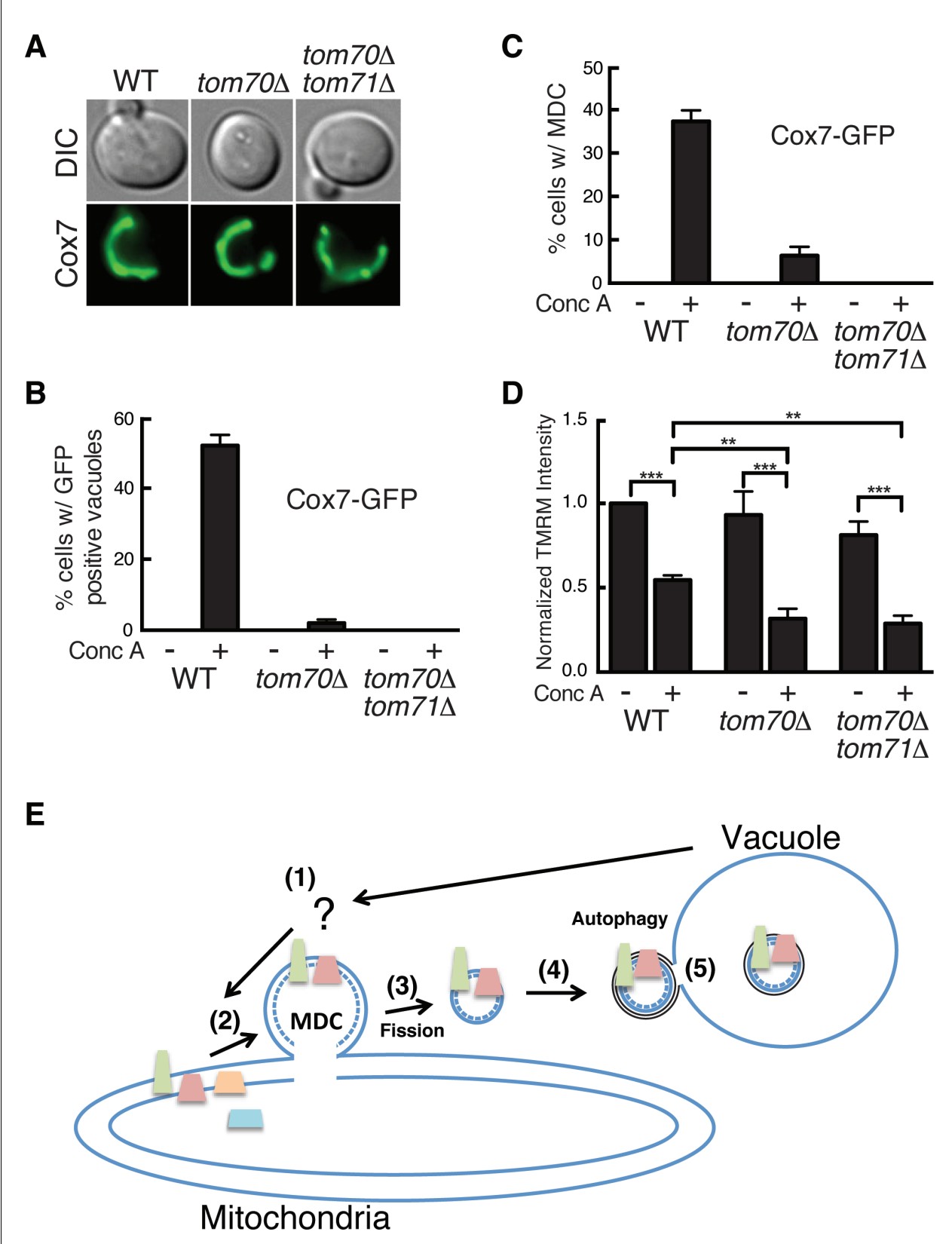

**Figure 7.** The mitochondrial import receptors Tom70 and Tom71 are required for MDC formation. (**A**) The MDC substrate Cox7 localizes to mitochondria lacking *TOM70* and *71*. Wild-type (WT) and the indicated mutant strains expressing the inner membrane protein Cox7-GFP were
*Figure 7 continued on next page*

Figure 7 continued

visualized by fluorescence microscopy. (B) Tom70 and 71 are required for vacuole delivery of Cox7-GFP. Quantification of Cox7-GFP in vacuoles of wild-type (WT) and the indicated mutant strains treated with concanamycin A (conc A) for 4 hr. Data represents percentage of cells with Cox7-GFP in the vacuole. Error bars represent standard deviation of 3 replicates. N = 100 for each replicate. (C) Tom70 and 71 are required for MDC formation. Quantification of Cox7-GFP containing MDCs in fission deficient strains lacking the indicated genes treated with concanamycin A (conc A) for 2 hr. Data represents percentage of cells with Cox7-GFP in MDCs. Error bars represent standard deviation of 3 replicates. N = 100 per replicate. (D) Failure to form MDC exacerbates membrane potential loss. Median fluorescence intensity of mitochondrial dye TMRM in wild-type (WT) and the indicated mutant strains treated with and without conc A for 4 hr as measured by flow cytometry. Median fluorescence intensity of a population of cells is presented as a percentage of the WT untreated sample (which is set at 100). N > 20,000 cells. Error bars represent standard deviation of three independent replicates. **p<0.01, ***p<0.001, multiple comparison one-way anova test. (E) Model of the MDC pathway. Loss of vacuole function caused by aging or other mechanisms produces an unknown signal (Step 1) that triggers MDC formation (Step 2). Select mitochondrial inner and outer membrane proteins are incorporated into MDCs, which are subsequently released from mitochondria by the fission GTPase Dnm1 (Step 3). MDCs are then engulfed by autophagosomes (Step 4), and delivered to the vacuole for degradation by autophagy (Step 5). It is currently not clear if the mitochondrial inner membrane (dashed line) is incorporated into MDCs.

The following figure supplement is available for figure 7:

**Figure supplement 1.** Further support for the role of Tom70 and 71 in MDC formation.

dysfunction. This pathway specifically destroys a subset of the mitochondrial membrane proteome through a series of steps outlined in *Figure 7E*. First, loss of acidity in the vacuole of aged or young cells triggers formation of a mitochondrial derived compartment, or MDC, through an unknown signal. Preexisting mitochondrial outer and inner membrane proteins that rely on Tom70 for import into the mitochondria are sequestered into the MDC through a mechanism that requires the import receptors Tom70/71. After formation, the entire MDC or portions of it are released from the mitochondria by a process that requires the mitochondrial fission machinery, and subsequently delivered to the vacuole by autophagy. Our data suggests that MDC formation helps protect mitochondria from vacuole-induced stress, as cells that cannot form MDCs exhibit a greater loss of membrane potential in response to disruption of vacuolar acidity (*Figure 7D*). A collapse of MDC-mediated autophagy occurs in very old yeast cells, leading to Dnm1-dependent formation of small Tom70-containing vesicle-like structures in the cytoplasm (*Figure 1A*). Interestingly, deletion of *DNM1* has been reported to extend lifespan in yeast (*Scheckhuber et al., 2007*), raising the possibility that these Tom70-containing mitochondrial fragments may contribute to cell toxicity or death in aged yeast.

The discovery of the MDC pathway raises new questions about the function of this system and how it relates to other known forms of mitochondrial protein degradation (*Anand et al., 2013*). The MDC pathway appears mechanistically distinct from other autophagy-dependent systems such as PINK1/Parkin- and Atg32-dependent mitophagy (*Kanki et al., 2011*; *Youle and Narendra, 2011*). Both of these pathways are thought to degrade whole portions of mitochondria, whereas MDCs selectively destroy a subset of mitochondrial membrane associated proteins. In this light, MDCs appear to have more in common with mitochondrial-derived vesicles, or MDVs, an autophagy-independent form of mitochondrial quality control thus far only identified in mammals (*Sugiura et al., 2014*). Several different classes of MDVs have been described to date, and all of them selectively incorporate a subset of the mitochondrial proteome for delivery to the peroxisome or lysosome (*Neuspiel et al., 2008*; *Soubannier et al., 2012a*). However, MDCs also differ from MDVs. Although the complete substrate specificity of MDVs is not well defined, it is clearly different than MDCs (*Neuspiel et al., 2008*; *Soubannier et al., 2012b*). Additionally, unlike MDCs, release of MDVs from mitochondria does not require the mitochondrial fission machinery (*Neuspiel et al., 2008*). Interestingly, recent studies have suggested that the PINK1/Parkin- and Atg32-mitophagy pathways might also act in substrate selective manners like MDCs and MDVs (*Abeliovich et al., 2013*; *Chan et al., 2011*; *Vincow et al., 2013*). This raises the intriguing possibility that substrate selectivity may be a common theme of many different mitochondrial protein degradation systems.

What is the function of the MDC pathway? The answer to this question may lie in the nature of the substrates targeted for degradation through MDCs. By screening through the mitochondrial proteome, we found that membrane proteins lacking defined mitochondrial-targeting presequences are sequestered into MDCs. These proteins require the mitochondrial surface receptor Tom70 for import into the mitochondria (*Schmidt et al., 2010*). The largest class of these Tom70-dependent

import substrates is the mitochondrial carrier proteins, an evolutionarily conserved family of nutrient transporters that facilitate exchange of metabolites across the mitochondrial inner membrane (*Palmieri and Pierri, 2010*). Yeast contain ~35 members of this family in their inner membrane, which have affinity for many different metabolites including amino acids, nucleotides, metals, and TCA cycle intermediates. Interestingly, MDC formation is triggered by inhibiting vacuolar function. One important function of the vacuole is storage of nutrients (*Klionsky et al., 1990*), and we previously showed that mitochondrial failure in response to loss of vacuole acidity results from impaired storage of nutrients such as amino acids within the vacuole lumen (*Hughes and Gottschling, 2012*). This raises the possibility that the purpose of the MDC pathway is to sequester mitochondrial nutrient transporters in response to cytoplasmic nutrient overload (*Wellen and Thompson, 2010*), perhaps to prevent unregulated nutrient influx into the mitochondria. Alternatively, the MDC pathway may degrade nutrient transporters to adjust mitochondrial metabolism towards a state preferable to survive vacuole impairment.

In considering the MDC in nutrient transporter destruction, it is worth noting the similarity of MDCs to a recently characterized vacuole-derived compartment which functions in the turnover of a vacuole nutrient transporter (*Li et al., 2015*). These compartments form from the vacuole membrane in response to cellular nutrient fluctuations, and have been shown to sequester a vacuolar lysine transporter away from the rest of the vacuolar membrane. They look morphologically similar to MDCs, but interestingly, are destroyed after release from the vacuole membrane by the multivesicular body pathway, not autophagy. Whether these vacuole-derived compartments are related in any way to MDCs is currently unclear. However, given the intimate metabolic relationship between the vacuole and mitochondria (*Rutter and Hughes, 2015*), and the recent identification of a physical tether (vCLAMP) between these organelles (*Elbaz-Alon et al., 2014*; *Honscher et al., 2014*), it will be important to explore the relationship between MDCs and vacuole-derived compartments (*Li et al., 2015*), as well as to determine the role of the MDC in vacuole-mitochondria crosstalk.

Another unanswered question is how are MDCs formed? At this point, we know that the MDC contains both inner and outer mitochondrial membrane-associated proteins, and that proteins incorporated into MDCs preexist within mitochondria. This suggests a sorting and segregation system exists in the inner and outer mitochondrial membranes. The nature of how that might occur is unclear, but the import receptor Tom70/71 plays a role in the formation of the MDC. This suggests that these proteins have roles beyond their function in mitochondrial import. Consistent with this idea, Tom70/71 are known to be required for recruiting an F-box protein Mfb1 to the mitochondrial surface (*Kondo-Okamoto et al., 2008*), and were recently shown to form an ER-mitochondrial tether with the ER sterol-transport protein Ltc1/Lam6 (*Murley et al., 2015*). It will ultimately be important to determine whether MDC's contain a single or double membrane, which along with identifying more MDC formation machinery will begin to shape our understanding of how proteins might be incorporated into these structures. At the moment, we cannot rule out the possibility that MDCs are formed through fusion of small mitochondrial-derived vesicles that are currently undetectable in our assay.

Finally, it will be interesting to determine how MDCs are linked to the autophagy machinery for degradation. Autophagy can selectively degrade a number of organelles, cellular substructures, and protein aggregates (*Stolz et al., 2014*). In mammals, the degradation of these structures is often facilitated by ubiquitination of target molecules to be degraded (*Kirkin et al., 2009*). Ubiquitin tags are recognized by adaptor proteins such as p62, which bind both ubiquitin and the autophagy protein Atg8/LC3 to link target molecules to the autophagy machinery (*Pankiv et al., 2007*). However, yeast homologs of p62 lack ubiquitin-binding domains, and until very recently, it was unclear if yeast could target proteins for autophagy-dependent degradation through ubiquitin tagging (*Rogov et al., 2014*). However, a recent study identified the conserved CUET protein Cue5 as an ubiquitin/Atg8-binding adaptor protein that facilitates autophagy-dependent turnover of ubiquitylated protein aggregates (*Lu et al., 2014*). It remains to be seen whether ubiquitylation and the CUET protein family play any role in the MDC pathway.

The discovery of the MDC pathway adds to a growing list of diverse systems for mitochondrial protein degradation. Understanding how these pathways function in concert with one another to maintain mitochondrial integrity in the face of various cellular stresses will ultimately be important for combating mitochondria-associated disease.

## Materials and methods

### Strains

All yeast strains are derivatives of *S. cerevisiae* S288c (*Brachmann et al., 1998*) and are listed in *Supplementary file 2*. Strains were created by one step PCR-mediated gene replacement and epitope tagging using standard techniques (*Brachmann et al., 1998*; *Sheff and Thorn, 2004*). Oligos to construct strains are listed in *Supplementary file 3*. Plasmid templates for tagging and knockout construction were from the previously described pRS, pKT, and pBSseries of vectors (*Brachmann et al., 1998*; *Shaner et al., 2004*; *Sheff and Thorn, 2004*). pBS34 was obtained from the Yeast Resource Center at the University of Washington with permission from Roger Tsien. RITE tagged strains were created using the previously described template plasmid pVL015 (*Verzijlbergen et al., 2010*). A collection of yeast strains expressing Tom70-mCherry/any protein-GFP was created by crossing a Tom70-mCherry query strain (UCC4997, see *Supplementary file 2*) to the yeast GFP strain collection (*Huh et al., 2003*) using a Biomek Robot and standard techniques for high-throughput strain construction (*Tong and Boone, 2006*). Strains were maintained and used for screening as diploids with both Tom70-mCherry and the GFP-fused proteins of interest in a heterozygous state. The final genotype of all strains in the collection is: MATa/MATα his3△1/his3△1 leu2△0/leu2△0 ura3△0/ura3△0 met15△0/+ lys2△0/+ anygene-GFP-His3MX/+ TOM70-mCherry-KanMX/+. Strains used in *Figure 5D* (Tcd2, Oac1, and Ilv2), as well as *Figure 5—figure supplement 1* (Sec63, Erg6, Pex11, Vrg4, and Sec7) and *Figure 7—figure supplement 1A–B* (Cox7) are from this strain collection. The GFP-ATG8 reporter strain used in *Figure 4—figure supplement 2* expresses an extra copy of *GFP-ATG8* from a *GPD* promoter integrated into an empty region of chromosome I (between 199456 and 199457). This strain was created by transformation and insertion of NotI-digested plasmid pAG306-GPD-eGFP-ATG8 chr I, which is described below.

### Plasmids

pAG306-GPD-eGFP-ATG8 chr I is a plasmid that can be integrated into an empty region of yeast chromosome I after digestion with the restriction enzyme NotI. The plasmid expresses *GFP-ATG8* from the constitutive *GPD* promoter. pAG306-GPD-eGFP-ATG8 chr I was constructed in two steps. First, we created pAG306-GPD-eGFP-ccdB chr I, a plasmid for high expression of N-terminal GFP fusion constructs from the *GPD* promoter that can be integrated into chromosome I (199456–199457) after NotI digestion. We generated pAG306-GPD-eGFP-ccdB chr I by ligation of a SmaI-digested fusion PCR product that contained two ~500 base pair regions of chromosome I flanking a NotI site into AatII-digested pAG306-GPD-eGFP-ccdB (Addgene plasmid 14308) (*Alberti et al., 2007*). We generated the fusion PCR product using oligos ChrI PartB SmaI F and ChrI PartA SmaI R to amplify two templates generated by PCR of yeast genomic DNA using oligo pairs ChrI PartA NotI F and ChrI PartA SmaI R, and ChrI PartB SmaI F and ChrI PartB NotI R, respectively. Second, we inserted *ATG8* into pAG306-GPD-eGFP-ccdB chr I from donor Gateway plasmid pDONR201-ATG8 (HIP ID ScCD00011665) using LR clonase according to manufacturer's instructions (Thermo Fisher Scientific, Waltham, MA) (*Hu et al., 2007*).

### Media and cell culture

As previously described (*Hughes and Gottschling, 2012*), cells were grown exponentially for 15 hr to a max density of $5 \times 10^6$ cells/ml before the start of all aging and MDC assays. This period of overnight log-phase growth was carried out to ensure vacuolar and mitochondrial uniformity across the cell population. Cells were cultured in YEPD (1% yeast extract, 2% peptone, 2% glucose) for all experiments. Yeast Complete (YC) medium used during construction of the Tom70-mCherry GFP strain collection was previously described (*Tong and Boone, 2006*; *van Leeuwen and Gottschling, 2002*). Concanamycin A (Sigma-Aldrich, St. Louis, MO) was added to cultures at a final concentration of 500 nM as indicated in figure legends. In the RITE tag experiments, cycloheximide (Sigma-Aldrich) was added at a final concentration of 50 μg/ml, and β-estradiol at 1 μM. In *Figure 3*, FCCP (Sigma-Aldrich), Antimycin A (Sigma-Aldrich), and hydrogen peroxide (Sigma-Aldrich) were added at to cultures at final concentrations of 10 μM, 20 μg/ml, and 3 mM, respectively.

## MDC Assays

For MDC assays, overnight log-phase cell cultures were grown in the presence or absence of conc A for the indicated time in the figure legends (0–6 hr). Cells were harvested by centrifugation, and imaged live in all experiments. The number of cells with MDCs or Tom70-GFP in vacuole-localized autophagosomes was quantified in each experiment at the appropriate timepoint. For screening of the Tom70-mCherry/GFP collection to identify MDC substrates, all strains were grown in batches of 20 following the same procedure used for all other MDC assays.

## Culturing and purification of aged MEP cells

For aging experiments, we used the Mother Enrichment Program (MEP) (*Lindstrom and Gottschling, 2009*) coupled to biotin/streptavidin purification to isolate cells of different replicative ages for microscopy analysis. Biotin labeling and purification of MEP cells was carried out exactly as previously described (*Hughes and Gottschling, 2012*). Briefly, to attach biotin to the cell surface, we washed $2.5 \times 10^7$ cells from a 15 hr YEPD log-phase culture twice in phosphate buffered saline, pH 7.4 (PBS) and resuspended in PBS with 3 mg/ml Sulfa-NHS-LC-Biotin (Thermo Fisher Scientific) at a final concentration of $2.5 \times 10^7$ cells/ml. Cells were incubated for 30 min at room temperature (RT), followed by two washes in PBS and one in YEPD. Biotinylated cells were resuspended in 10 ml of YEPD at $2.5 \times 10^6$ cells/ml and recovered with shaking for 2 hr at 30°C. These cells were used to seed cultures at a density of $2 \times 10^4$ biotinylated cells/ml in YEPD for aging experiments. To initiate the MEP aging program, β-estradiol (1 μM) was added to cultures and cells were grown at 30° for an appropriate time to obtain cells of a desired age (~1 hr for young cells, 12 hr for middle-aged, and 24 hr for old cells). Cell densities never exceeded $4 \times 10^6$ cells/ml. $1 \times 10^8$ total cells were harvested for purification and microscopy analysis at each timepoint.

For purification after aging, cells were washed twice with PBS, and then resuspended in 500 μl of PBS at a density of $2 \times 10^8$ cells/ml. Cells were then incubated for 30 min at RT with 25 μl of streptavidin-coated magnetic beads (MicroMACS, Miltenyi Biotec, Germany). Cells were then washed twice in PBS, resuspended in 8 ml of PBS, and loaded onto a LS MACS column (Miltenyi Biotec) that had been equilibrated with 5 ml of PBS. Cells on the column were washed twice with 8 ml of PBS. Columns were then removed from the magnetic field and aged cells were eluted by gravity flow with 8 ml of PBS. Cells were centrifuged to concentrate them for microscopy analysis.

## Fluorescent staining

3,3′-dihexyloxacarbocyanine iodide (DiOC$_6$) (Thermo Fisher Scientific) staining was carried out as previously described (*Hughes and Gottschling, 2012*). Briefly, $2 \times 10^6$ log-phase cells were washed once in 10 mM HEPES, pH 7.6 + 5% glucose and then resuspended in 1 ml of the same buffer containing 175 nM DiOC$_6$. Cells were then incubated for 15 min at RT, followed by two washes with 10 mM HEPES, pH 7.6 + 5% glucose. Cells were resuspended in 10 mM HEPES, pH 7.6 + 5% glucose for imaging. Tetramethylrhodamine methyl ester (TMRM) (Thermo Fisher Scientific) staining was carried out exactly as DiOC$_6$ staining, except that cells were incubated with 50 nM TMRM.

For aging experiments, cell age was determined by calcofluor (Sigma-Aldrich) staining of bud scars. For this analysis, 5 μg/ml calcofluor was included in the first post-staining wash step prior to imaging. For each experiment, cells were grouped into 3 categories based on age range: Young (0–4 budscars); middle-aged (7–12); and old (>17).

## Microscopy

For fluorescence microscopy analysis in all figures except those noted below, cells were visualized under 60X oil magnification using a Nikon Eclipse E800 with the appropriate filter set. Images were acquired with a CoolSNAP HQ$^2$ CCD camera (Photometrics, Tucson, AZ) and quantified and processed using Metamorph version 7.1.1.0 imaging software. For DiOC$_6$ experiments, cells were scored as reduced if they exhibited at least a 2-fold decrease in mean fluorescence intensity. Microscopy analysis in *Figures 3*, *7*, and *Figure 4—figure supplements 1B* and 2 was carried out using a 100X oil objective on a Zeiss AxioImager M2 using the appropriate filter sets. Images were acquired with an Axiocam 506 mono camera, and processed using Zen imaging software (Zeiss, Germany). Images in *Figure 4—figure supplements 1B* and *2* represent maximum intensity projections of 6–10 step Z-stacks.

For *Videos 1–3* showing Tom70-GFP and Vph1-mCherry in old yeast cells, Z-stack images were acquired with a DeltaVision Elite imaging system (GE Healthcare, United Kingdom) using a 100X, 1.4 NA oil immersion lens and a CoolSNAP HQ$^2$ CCD camera (Photometrics). Images were deconvolved with SoftWoRx 6.5.2 image analysis software (GE Healthcare) and images of the vacuole and mitochondria were IsoSurface rendered in Imaris 8.2.0 software (Bitplane, Switzerland).

### Flow cytometry

After DiOC$_6$ or TMRM staining, cells were analyzed on a BD LSRFortessa X-20 equipped with the appropriate filter sets. At least 10,000 events were analyzed for each sample. Statistical analysis for *Figure 7D* was conducted using Graphpad Prism software.

### Protein preparation and immunoblotting

$2 \times 10^7$ log-phase cells were resuspended in 100 µl H$_2$O. An equal volume of 0.2 M NaOH was added to the cell suspension, and cells were incubated 5 min at room temperature. Samples were then centrifuged at 20,000 x g for 10 min at 4°C. Pellets were resuspended in SDS-lysis buffer (10 mM Tris-HCl, pH 6.8, 100 mM NaCl, 1% SDS, 1 mM EDTA, and 1 mM EGTA) containing protease inhibitors (leupeptin, pepstatin, PMSF, and aprotinin) for western blot analysis. Immunoblotting was carried out exactly as previously described (*Hughes and Gottschling, 2012*). Anti-GFP primary antibody was from Roche (Switzerland) (#11814460001), and secondary HRP-conjugated antibodies from Jackson Immunoresearch (West Grove, PA).

## Acknowledgements

We thank members of the Hughes and Gottschling labs for helpful discussions and their critical review of the manuscript; Janet Shaw (Utah) and Jared Rutter (Utah) for helpful discussions; and Julio Vazquez and David McDonald (FHCRC Scientific Imaging) for assistance with imaging and software analysis for *Videos 1–3*. This work was supported by the Searle Scholars Program (ALH) and NIH grants AG043095 (ALH), AG037512 (DEG), and AG023779 (DEG).

## Additional information

### Funding

| Funder | Grant reference number | Author |
| --- | --- | --- |
| National Institute on Aging | AG043095 | Adam L Hughes |
| Kinship Foundation | Searle Scholars Award | Adam L Hughes |
| National Institute on Aging | AG037512 | Daniel E Gottschling |
| National Institute on Aging | AG023779 | Daniel E Gottschling |

The funders had no role in study design, data collection and interpretation, or the decision to submit the work for publication.

### Author contributions

ALH, Conception and design, Acquisition of data, Analysis and interpretation of data, Drafting or revising the article; CEH, KAH, Acquisition of data, Analysis and interpretation of data; NY, Acquisition of data, Analysis and interpretation of data, Contributed unpublished essential data or reagents; DEG, Conception and design, Analysis and interpretation of data, Drafting or revising the article

### Author ORCIDs

Adam L Hughes, http://orcid.org/0000-0002-7095-3793
Daniel E Gottschling, http://orcid.org/0000-0002-7303-6552

## Additional files

**Supplementary files**
• Supplementary file 1. Complete results of MDC screen.

• Supplementary file 2. Yeast strains used in this study.

• Supplementary file 3. Oligos used in this study.

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
