## [Decision Letter]

Thank you for submitting your work entitled "Selective sorting and destruction of mitochondrial membrane proteins in aged yeast" for consideration by *eLife*. Your article has been favorably evaluated by Richard Losick (Senior editor) and four reviewers, one of whom, Andrew Dillin, is a member of our Board of Reviewing Editors.

The following individuals involved in review of your submission have agreed to reveal their identity: Suzanne Pfeffer, Thomas Nyström, and Liza Pon (peer reviewers).

The reviewers have discussed the reviews with one another and the Reviewing Editor has drafted this decision to help you prepare a revised submission.

As you will find, all three reviewers and the Reviewing editor are enthusiastic about the body of work presented. The authors put forth very compelling and interesting data to suggest that mitochondria in yeast might possess a quality control system to selectively remove outer membrane proteins in a vesicle like structure that is ultimately destined for the lysosome. The idea of a mitochondrial derived compartment that plays a role in mitochondrial integrity is exciting and provides an explanation for selective degradation of some mitochondrial clients.

While there are differences among some of the reviewers’ comments, a shared theme of all includes better characterization of the MDC. This is particularly relevant since this is the first description of this compartment and the similarities/differences of mitochondrial derived vesicles (MDVs) needs to be better flushed out. Along these lines, the origination of the MDC with the autophagy machinery needs a deeper look and each reviewer has suggested a few experiments. Finally, the analysis of which mitochondrial perturbations can generate the MDC needs a little more analysis and suggestions to include mitochondrial toxins is a good place to start. Overall, the reviews are very favorable and please contact the Reviewing editor if you need additional input.

*Reviewer #1:*

In the manuscript submitted by Hughes et al. entitled "Selective sorting and destruction of mitochondrial membrane proteins in aged yeast", the authors put forth very compelling and interesting data to suggest that mitochondria in yeast might possess a quality control system to selectively remove outer membrane proteins in a vesicle like structure that is ultimately destined for the lysosome.

The data to support such events include:

1) Tom70-GFP is found inside the vacuole (lacking the major peptidase, pep4) of old cells and conditions where the vacuole acidity is altered (conc A treated cells).

2) The trafficking to the vacuole appears partially dependent upon the autophagy pathway since it is *atg5* dependent.

3) Trafficking to the vacuole is independent of mitophagy. Experiment shows Tom70-GFP in vacuole in *atg32* mutant cells.

4) The origin of the vesicle is suggested to require the fission machinery since Dnm1 mutations block Tom70-GFP accumulation in vacuole.

5) During aging, the vacuole becomes fragments and the appearance of small vesicles containing Tom70-GFP is found.

6) Tom70-GFP vesicles appear shortly after depolarization of mitochondria before entry to vacuole.

7) Cargo is selective to outer membrane proteins that require Tom70/71. Tom20 not required.

8) Trafficking to the vacuole seems to include both newly synthesized and older proteins, suggesting it is not a back-up of non-translocated proteins.

In summary, the paper is very fascinating and suggests a new avenue for mitochondrial quality control that could be distinct from mitophagy and mitochondrial derived vesicles (MDVs) found in metazoan cells.

To compare and contrast MDVs from MDC:

Cargo for MDVs include oxidized proteins, lipids, outer membrane proteins and the target is lysosomes and peroxisomes. MDVs require PINK1/Parkin, DRP-1 independent and autophagy independent. The stress that causes MDVs to form include antimycin and other harsh mitochondrial toxins.

The cargo of MDCs contains outer membrane proteins and import carriers that are targeted to the lysosome. Dnm1 dependent and autophagy independent. Yeast do not have a PINK1/Parkin pathway, so it is unknown if they could be involved if the MDC is conserved in metazoa. The aging process and de-acidification of the lysosome allows observation of MDCs.

The major questions to address form this work:

1) While the RITE experiment is very clever and suggests that newly made and older proteins can be found in MDCs (well, actually found in lysosome), how sure are the authors that fused proteins of interest are not sitting in the cytoplasm due to the inability to be translocated to the mitochondria? Because all perturbations tested resulted in membrane depolarization and the Tom70/71 complex is essential for this process, which requires membrane polarization for import, could this be a clearance mechanism for non-transported proteins? For example, could a cytoplasmic localized Tom70-GFP be found in the lysosome using the conditions described here?

2) Do other mitochondrial stresses also induce MDC? Oxidative stress or starvation?

3) The *ATG32* experiments are very interesting. However, my understanding of the literature is that mitophagy mediated by *ATG32* only occurs during starvation. It is not clear if the negative results showing that *ATG32* is not required were done under vegetative or starved conditions where *ATG3*2 is required.

4) Because there is close correlation of "fizzed" mitochondria with appearance of the MDC, does blocking fusion give the same result?

5) The authors correctly claim that the MDC travels to the vacuole because they can see the cargo accumulate there in a *pep4* deletions strain. However, without having a similar tool to see the trapped cargo (i.e. a peptidase knockout for each potential compartment) it might be premature to rule out that the MDCs do not travel to other compartments or might even fuse with the plasma membrane and allow deposition of cargo to the outside of the cell.

6) The idea of the MDC is very compelling, but EM data to describe its structure would be more informative (i.e. does it have a membrane?).

*Reviewer #2:*

This is a very well written and very interesting paper that describes a new pathway for the turnover of mitochondrial nutrient transporters that is distinct from classic mitophagy and may resemble the formation of mitochondrial derived vesicles seen in mammalian cells by McBride and coworkers – however this pathway requires mitochondrial fission. The authors have identified a select group of Tom70/Tom71 import substrates that are targeted to the vacuole upon loss of vacuole acidification or loss of mitochondrial membrane potential. Using a clever tagging approach they show that already synthesized and apparently translocated proteins participate in this process, implying that they may be organized by Tom70/71 after synthesis. The data are of high quality and the findings of broad interest that readers of *eLife* will very much appreciate.

Comments.

1) The authors refer to this pathway as a quality control pathway that degrades mitochondrial components from aged/damaged mitochondria, but since it also is induced upon loss of vacuole acidification, couldn't it just as well be described as a pathway to regulate mitochondrial function when nutrients are less available (and metabolic need is less) that is distinct from the Atg32 pathway? Please discuss and consider modifying text/Abstract/impact statement

2) Please comment on whether the levels of Tom70/71 and/or their substrates change upon vacuole neutralization or mitochondrial membrane depolarization.

Overall, well done and so nicely presented such that non-experts can appreciate it.

*Reviewer #3:*

The authors identified a mechanism for mitochondrial outer membrane proteins and mitochondrial carrier proteins from the inner membrane are selectively removed from mitochondria in old cells and in all cells in response to loss of vacuolar acidity. Protein removal occurs by DNM1- and Tom70/71-dependent formation of a mitochondria derived compartment (MDC) and *ATG5*-dependent but *ATG32*-independent degradation of MDCs by the vacuole. The authors also provide evidence that MDCs are distinct from MDV.

The work is novel and of clear interest to the field. Major concerns regarding this work are outlined below.

1) The resolution of fragmented vacuoles and Tom70-GFP in Figure 1 is low. On the basis of the image shown, it is not clear that Tom70-GFP-containing punctate is excluded from fragmented vacuoles in old cells. Is there a decrease in *PEP4*-dependent degradation of Tom70-GFP to free GFP in old compared to middle aged cells?

2) The authors find that *DNM1* is required for MDC formation. If the authors want to make the claim that MDCs are released from mitochondria by Dnm1-dependent mitochondrial fission, they need to provide evidence for a direct role of Dnm1 in MDC formation. For example, does Dnm1 localize to MDCs as they bud from mitochondria?

3) The authors also find that the *ATG5* is required for degradation of MDCs by the vacuole, and propose that MDC degradation occurs by macroautophagy. If they want to make the claim that MDCs are enveloped in autophagosomes prior to degradation in the vacuole, they need to test whether autophagosome marker proteins are associated with MDCs and whether deletion of *DNM1* affects this localization.

4) The finding that select proteins are excluded from MDCs in conc A-treated cells is interesting and important.

A) Since mitochondria are generally more fragmented in old yeast compared to young yeast, it is not clear that the punctate Tom70-GFP-containing structures observed in old yeast are MDCs. Is Tim50 excluded from Tom70-GFP-containing structures in old yeast?

B) To confirm that there is degradation of a select set of proteins in MDCs, it would be interesting to see if Tim50 is not degraded by autophagy in conc A-treated cells.

5) It is not clear whether *tom70/71∆* cells are more sensitive to conc A because they have defects in protein import or defects in MDC formation. The text should be revised accordingly.

6) Although loss of vacuolar acidification does result in a decline in mitochondria membrane potential, it is not clear that the proteins that are removed in MDCs are dysfunctional. In fact, the authors speculate that MDC-dependent remodeling of mitochondria alters mitochondrial function in metabolite transport. Therefore, it is possible that the MDC pathway is not a quality control pathway, but a pathway for control of mitochondrial function. The manuscript would benefit from studies to distinguish between these two possibilities. For example, is loss of mitochondrial membrane potential or function sufficient to trigger MDC formation? Either outcome would be of interest to the field.

*Reviewer #4:*

In this paper, the authors report on the discovery of a new mitochondrial quality control system operating in yeast. They first show, taking advantage of *pep4* mutant, that the mitochondrial protein Tom70 is degraded in old yeast cells by the vacuole in an Atg5-dependent manner indicating a role for autophagy in the delivery of Tom70 the vacuole. In addition, the delivery of Tom70 was abrogated in *dnm1* mutant cells suggestion that fission from mitochondria is required for proper rerouting of Tom70 to degradation. It is further shown that loss of vacuolar acidity, which the authors have shown previously occurs early during mother cell aging and causes mitochondrial dysfunction, triggers the destruction of Tom70 via the route discovered. Prior to delivery to the vacuole, Tom70 congregated into large foci, named mitochondria-derived compartment (MDC) and this assembly of Tom70 was independent on both Dnm1 and Atg5.

Most interestingly, MDC was then demonstrated to be specific for mitochondrial proteins lacking the cleavable presequence and the authors show that preexisting mitochondrial proteins (along with newly made ones) are targeted to MDC in a Tom70-dependent manner.

This is a very interesting and well-written paper, which reports on a new concept of mitochondrial quality control in yeast. As the authors themselves point out, there are several issues that remain to be elucidated, for example the mechanisms behind MDC formation and its specificity, how MDCs are mechanistically transferred to the vacuole, and the physiological significance of MDC in yeast aging/rejuvenation. Nevertheless, the paper entails a very important discovery and the experiments have taken this discovery to a level that warrants publication. I have a few comments and questions that the authors might want to consider for further clarifying some points of the paper:

1) Concerning the requirement for fission. It is clear from the data that *Fis1* cannot compensate for the lack of Dnm1 as the *dnm1* deletion completely blocked Tom70 delivery to the vacuole. Nevertheless, it would be interesting to see if a *fis1* deletion have some kind of defect also as *Fis1* is required for proper localization of Dnm1. Also, as *dnm1* deletion mutants display a defect in endomembrane trafficking one wonders if such effects could be part of the problem in delivering MDCs to the vacuole? Cannot come up with a simple genetic way/experiment to separate mitochondrial fission defects form endomembrane trafficking defects but the authors could perhaps comment on this.

2) In the experiments using conc-A to trigger loss of vacuolar acidity, the delivery of Tom70 to the vacuole is interesting for several reasons and one wonders if this could be discussed a bit further. First, loss of vacuolar acidity is effectively blocking endomembrane trafficking to the vacuole. Does this mean that the Atg5-dependet route is "immune" to such pH effects in the vacuole or does it indicate that MDCs are transferred from the mitochondria to the vacuole by a more direct route: mitochondrial-vacuolar contacts?

The other interesting possibility (as the phenomenon is seen in old cells and during conc-A treatment) is that a loss of vacuolar acidity is, in fact, required for triggering the MDCs formation and delivery. Could it be tested perhaps if drugs affecting mitochondrial function but leaving the pH of vacuole intact can trigger MDC formation and delivery? In other words, approach the generality of the system discovered upon mitochondrial dysfunction.

3) It is stated that the formation of small vesicle-like structures (Tom70 associated) in the cytoplasm of old cells require mitochondrial fission (Dnm1) indicating that these membrane structures are derived from mitochondria. It is very hard to see these structures in the picture of the *dnm1* strain – perhaps a better version could be added? Also, as the experiment uses Vph1 as a reporter, does not this indicate that the vesicle-like structures are derived from the vacuole (vacuolar fragmentation?) rather than formed by fission from mitochondria? At present, it is a bit unclear what these structures are, how they were formed, and how they relate to the MDC route discovered.

4) The authors suggest that MDC formation helps protect mitochondria from vacuole-induced stress as a greater loss in membrane potential is seen in tom70/71 mutants subjected to conc A treatment. Is it possible that this loss of potential in tom70/71 mutants could be related to other shortcomings than the formation of MDC? Also, did *atg5* and *dnm1*, but not *atg32*, mutants display a similar loss of potential upon conc-A treatment?

5) Since aging is the entry port in this paper, I wonder if the authors could discuss briefly the possible significance of MDC in yeast longevity? As *dnm1* and *fis1* deletion mutants display a somewhat longer replicative lifespan, is it possible that a reduction in mitochondrial fragmentation (positive effect) in these mutants offsets the negative effect of an abrogated MDC response? Is the lifespan on *tom70, atg5, atg32* mutants known?

---

## [Author Response]

Reviewer #1:

1) While the RITE experiment is very clever and suggests that newly made and older proteins can be found in MDCs (well, actually found in lysosome), how sure are the authors that fused proteins of interest are not sitting in the cytoplasm due to the inability to be translocated to the mitochondria? Because all perturbations tested resulted in membrane depolarization and the Tom70/71 complex is essential for this process, which requires membrane polarization for import, could this be a clearance mechanism for non-transported proteins? For example, could a cytoplasmic localized Tom70-GFP be found in the lysosome using the conditions described here?

We appreciate the reviewer’s concerns that the MDC may be a collection of newly synthesized, unimported proteins, and we specifically discuss this possibility in the paper. However, several pieces of data in the manuscript strongly support the conclusion that the MDC degrades preexisting mitochondrial proteins. The first is the inclusion of cycloheximide in combination with the RITE tag experiment in Figure 6. As a control to address this point, we previously showed that MDCs still form in the presence of cycloheximide, which completely blocks all new synthesis of the RITE tag protein. Thus, the MDC is not formed from newly synthesized, unimported protein in this instance. In addition, outer membrane proteins that are MDC substrates do not require mitochondrial membrane potential for import into the mitochondria, and would not be expected to accumulate outside the mitochondrial under depolarizing conditions. Lastly, we have now included data in the manuscript as new Figure 3 (see question 2 below) that shows that loss of membrane potential is not sufficient to trigger MDC formation. If the MDC were simply a collection of unimported substrates, it would form under all mitochondrial depolarizing conditions that block mitochondrial import (e.g. loss of membrane potential), but we find that it does not.

2) Do other mitochondrial stresses also induce MDC? Oxidative stress or starvation?

In response to the reviewers’ suggestion, we tested whether mitochondrial depolarizing agents FCCP and Antimycin A could induce MDC formation. We also tested hydrogen peroxide and amino acid starvation as well. All four treatments failed to induced MDC formation, suggesting the MDC is triggered by an unidentified signal originating from loss of vacuole acidity. The results from the FCCP, Antimycin A, and hydrogen peroxide experiments showing their impacts on mitochondrial membrane potential and structure but failure to trigger MDCs have been included in the manuscript as new Figure 3.

3) The ATG32 experiments are very interesting. However, my understanding of the literature is that mitophagy mediated by ATG32 only occurs during starvation. It is not clear if the negative results showing that ATG32 is not required were done under vegetative or starved conditions where ATG32 is required.

The experiments addressing the role of *ATG32* in the process were conducted in old cells or cells treated with concanamycin A to induce MDC formation, which are not conditions in which *ATG32* has been previously implicated or tested. Our results show that *ATG32* is not required for MDC formation, and the point of including this mutant is to show that the MDC pathway is distinct from the previously characterized Atg32-dependent mitochondrial autophagy pathway.

4) Because there is close correlation of "fizzed" mitochondria with appearance of the MDC, does blocking fusion give the same result?

In response to the reviewer’s suggestion, we performed experiments testing whether MDCs are constitutively formed in mitochondrial fusion mutants. We found that deletion of the fusion protein *FZO1* does not cause MDC formation. However, the interpretation of this experiment is not straightforward, because loss of *FZO1* causes yeast cells to lose mitochondrial DNA. Loss of mitochondrial DNA has previously been shown to interfere with vacuole-mitochondrial signaling (PMID: 23502676). Thus, although our results were negative, they may be complicated by the lack of mitochondrial DNA. Because of this, we did not include this data in the revised manuscript.

5) The authors correctly claim that the MDC travels to the vacuole because they can see the cargo accumulate there in a pep4 deletions strain. However, without having a similar tool to see the trapped cargo (i.e. a peptidase knockout for each potential compartment) it might be premature to rule out that the MDCs do not travel to other compartments or might even fuse with the plasma membrane and allow deposition of cargo to the outside of the cell.

We agree with the reviewer that there may be other fates of MDCs beyond the vacuole that we cannot visualize at this time. If other fates exist, they may be revealed in longer term studies. The text of the manuscript has been modified to reflect this uncertainty.

6) The idea of the MDC is very compelling, but EM data to describe its structure would be more informative (i.e. does it have a membrane?).

We agree with the reviewer that EM data describing the MDC will be very informative. However, these studies are non-trivial to perform and while ongoing, have yet to be achieved. We believe that the present study already provides a great deal of support for an unexpected discovery as presented.

Reviewer #2:

*1) The authors refer to this pathway as a quality control pathway that degrades mitochondrial components from aged/damaged mitochondria, but since it also is induced upon loss of vacuole acidification, couldn't it just as well be described as a pathway to regulate mitochondrial function when nutrients are less available (and metabolic need is less) that is distinct from the Atg32 pathway? Please discuss and consider modifying text/Abstract/impact statement.*

We agree with the reviewer that the MDC pathway may not necessarily be a quality control pathway, but rather, a pathway involved in metabolic homeostasis. With the current set of data, we cannot exclude either possibility. As suggested, we have gone through the manuscript and modified our reference to the MDC pathway as a quality control pathway, and instead discuss it as a protein degradation pathway that responds to changes in the pH of the vacuole and modifies the mitochondrial proteome for reasons that remain to be determined.

*2) Please comment on whether the levels of Tom70/71 and/or their substrates change upon vacuole neutralization or mitochondrial membrane depolarization.*

We would expect that the levels of these proteins would change under conditions of MDC activation because they are targeted for degradation and removed from mitochondria. However, as can be seen in the western blot in Figure 2, the protein level of Tom70 (as an example) does not change. This lack of change in protein levels likely results from the fact that concanamycin A is used to induce this pathway, which ultimately also blocks degradation of the autophagosomes containing this protein in the vacuole (Nakamura et al., J of Biochem 1997). Thus, the total protein levels do not significantly change within cells, but we would suspect that levels of proteins within the mitochondria itself decline. It is not clear that this would occur with mitochondrial membrane depolarization, because as we now show in new Figure 3, loss of mitochondrial membrane potential alone is not sufficient to trigger MDC formation.

Reviewer #3:

1) The resolution of fragmented vacuoles and Tom70-GFP in Figure 1 is low. On the basis of the image shown, it is not clear that Tom70-GFP-containing punctate are excluded from fragmented vacuoles in old cells. Is there a decrease in PEP4-dependent degradation of Tom70-GFP to free GFP in old compared to middle aged cells?

Because the level of protein delivery to the vacuole in old cells is so low, we are unable to determine the Pep4-dependent degradation of Tom70-GFP by western blot as we conducted in young cells treated with conc A. However, we agree with the reviewer that it is difficult to ascertain in Figure 1 whether the small Tom70-fragments are within vacuoles or not in old cells. To address this issue, we have now included Video 1–Video 3, which show 3D reconstructions of old WT, *atg5△*, and *dnm1*△ cells taken with a DeltaVision microscope showing mitochondria (Tom70-GFP) and vacuole (Vph1-mCherry). As can be seen in the Video 1, the small Tom70-GFP punctae are outside of the fragmented vacuoles. These punctae are also present in *atg5△* cells (Video 2), and are not formed in *dnm1*△ cells (Video 3).

*2) The authors find that DNM1 is required for MDC formation. If the authors want to make the claim that MDCs are released from mitochondria by Dnm1-dependent mitochondrial fission, they need to provide evidence for a direct role of Dnm1 in MDC formation. For example, does Dnm1 localize to MDCs as they bud from mitochondria?*

At the reviewer’s suggestion, we tested whether Dnm1-GFP colocalizes with Tom70-mCherry labeled MDCs, and found that Dnm1-GFP foci localize at or around MDCs in 88% of cells. This new data is included as Figure 4—figure supplement 1.

3) The authors also find that the ATG5 is required for degradation of MDCs by the vacuole, and propose that MDC degradation occurs by macroautophagy. If they want to make the claim that MDCs are enveloped in autophagosomes prior to degradation in the vacuole, they need to test whether autophagosome marker proteins are associated with MDCs and whether deletion of DNM1 affects this localization.

In response to the reviewer’s suggestion, we tested whether Tom70-mCherry-labeled MDCs colocalize with the classic autophagy marker GFP-Atg8. GFP-Atg8 localizes to the pre-autophagosomal structure (PAS) in yeast, which is near the vacuole surface and marks sites of autophagosome formation. We found that the majority (68%) of MDCs localize at or near the PAS. Those that did not were away from the vacuole surface. These results suggest that MDCs form at the mitochondria away from vacuoles, and then are recruited to the PAS to be engulfed by autophagy. This data has been included in the revised manuscript as Figure 4—figure supplement 2.

*4) The finding that select proteins are excluded from MDCs in Conc A-treated cells is interesting and important.*

A) Since mitochondria are generally more fragmented in old yeast compared to young yeast, it is not clear that the punctate Tom70-GFP-containing structures observed in old yeast are MDCs. Is Tim50 excluded from Tom70-GFP-containing structures in old yeast?

We agree with the reviewer’s comment that MDC substrate selectivity is important, and indeed in our previous version of the manuscript, we do in fact show that Tim50-GFP is excluded from the MDC (marked with Tom70-mCherry) in aged cells (Figure 4 in original manuscript, now Figure 5 in the revised manuscript).

B) To confirm that there is degradation of a select set of proteins in MDCs, it would be interesting to see if Tim50 is not degraded by autophagy in Conc A-treated cells.

We agree with the reviewer’s suggestion, and have now included western blot data showing that unlike Tom70, Tim50 is not processed by Pep4 in conc A-treated cells. This new data is included as Figure 5.

5) It is not clear whether tom70/71∆ cells are more sensitive to Conc A because they have defects in protein import or defects in MDC formation. The text should be revised accordingly.

As the reviewer suggested, we added text to this section of the manuscript suggesting that the enhanced sensitivity to conc A in these mutants may be unrelated to their function in the MDC pathway.

6) Although loss of vacuolar acidification does result in a decline in mitochondria membrane potential, it is not clear that the proteins that are removed in MDCs are dysfunctional. In fact, the authors speculate that MDC-dependent remodeling of mitochondria alters mitochondrial function in metabolite transport. Therefore, it is possible that the MDC pathway is not a quality control pathway, but a pathway for control of mitochondrial function. The manuscript would benefit from studies to distinguish between these two possibilities. For example, is loss of mitochondrial membrane potential or function sufficient to trigger MDC formation? Either outcome would be of interest to the field.

Reviewers 1 and 2 raised the same important points, and as suggested, we addressed these issues both experimentally and by changing the text throughout the manuscript. Please see our detailed response to these suggestions in Reviewer 1’s section, question 2, and reviewer 2’s section, question 1. Briefly, we added a new Figure 3 to the manuscript addressing whether loss of mitochondrial membrane potential or oxidative stress is sufficient to trigger MDC formation by treating cells with FCCP, antimycin A, or hydrogen peroxide. None of these three treatments resulted in MDC formation, suggesting that loss of membrane potential is not sufficient to trigger this pathway. We also agree with the reviewer(s) that the MDC pathway may not necessarily be a quality control pathway, but rather, a pathway for another type of mitochondrial dynamics/reprogramming. With the current set of data, we cannot exclude either possibility. As noted above, we have modified the manuscript accordingly to reflect this different view. The signal that activates the MDC pathway in response to changes in vacuole function is unknown, but could very well be metabolic in nature.

Reviewer #4:

*1) Concerning the requirement for fission. It is clear from the data that Fis1 cannot compensate for the lack of Dnm1 as the dnm1 deletion completely blocked Tom70 delivery to the vacuole. Nevertheless, it would be interesting to see if a fis1 deletion have some kind of defect also as Fis1 is required for proper localization of Dnm1. Also, as dnm1 deletion mutants display a defect in endomembrane trafficking one wonders if such effects could be part of the problem in delivering MDCs to the vacuole? Cannot come up with a simple genetic way/experiment to separate mitochondrial fission defects form endomembrane trafficking defects but the authors could perhaps comment on this.*

At the reviewer’s suggestion, we examined the MDC pathway in strains lacking *FIS1*, and found that loss of *FIS1* blocks MDC release from mitochondria and subsequent delivery to the vacuole, exactly as previously observed in cells lacking *DNM1*. This new data has been added to the paper as part of Figure 4—figure supplement 1. As the reviewer suggests, it is certainly possible that Dnm1 could play a role beyond mitochondrial fission in trafficking of the MDC to the vacuole. However, we feel that our new data showing that *FIS1* is also required (which phenocopies loss of *DNM1* and is not known to play a role in other endomembrane trafficking reactions), and that Dnm1-GFP foci localize around the MDC, strengthens the argument that fission is required for release of MDCs from the mitochondria. Thus, we chose not to include an extra statement regarding a potential separate role for Dnm1 in this process.

2) In the experiments using conc-A to trigger loss of vacuolar acidity, the delivery of Tom70 to the vacuole is interesting for several reasons and one wonders if this could be discussed a bit further. First, loss of vacuolar acidity is effectively blocking endomembrane trafficking to the vacuole. Does this mean that the Atg5-dependet route is "immune" to such pH effects in the vacuole or does it indicate that MDCs are transferred from the mitochondria to the vacuole by a more direct route: mitochondrial-vacuolar contacts?

We share the reviewer’s view that the role of vacuolar acidity in this process is very interesting, and that the mechanism of exactly how the MDC gets transferred from the mitochondria to the vacuole isn’t completely clear (most specifically, how MDCs get linked to the autophagy machinery). Our current data suggests that the MDC or pieces of it gets released from the mitochondria by fission, and then degraded by autophagy in autophagosomes. This suggests that MDCs follow a traditional autophagy degradation route to the vacuole through autophagosomes. In support of this, we added data to the paper showing that loss of *VAM3* prevents the appearance of Tom70 within the vacuole of conc A treated cells. Vam3 is part of the vacuolar homotypic fusion machinery, and is required for fusion of autophagosomes with vacuoles. This data has been added to Figure 2 and Figure 4, and strongly supports the conclusion that MDCs are engulfed by autophagosomes and delivered to the vacuole through the normal autophagy route. The reviewer raises the idea that conc A should interfere with endomembrane trafficking to the vacuole. While this is the case in mammals, this is not true in yeast. We added a brief discussion of this point in the paper, highlighting previous literature that shows that autophagosomes are delivered efficiently to vacuoles in the absence of vacuolar acidity (Nakamura et al., J of Biochem 1997). However, autophagosomes cannot be broken down in non-acidic vacuoles and accumulate there, and we see this in our assay as well.

The other interesting possibility (as the phenomenon is seen in old cells and during conc-A treatment) is that a loss of vacuolar acidity is, in fact, required for triggering the MDCs formation and delivery.

Yes, our current data supports the idea that loss of vacuolar acidity actually triggers formation of the MDC. This is most clearly illustrated in Figure 2 and Figure 4. In these assays, we show that blocking protein degradation alone, through loss of *PEP4,* does not causes accumulation of Tom70-GFP in the vacuole of young cells, nor does it cause formation of the MDC. Instead, the addition of conc A to these cells triggers both the formation of the MDC and accumulation of Tom70-GFP in the vacuole. Thus, this pathway is not constitutively on in young cells, and must be induced by a signal originating from loss of vacuolar acidity. We have added a sentence further clarifying this point in the revised manuscript.

Could it be tested perhaps if drugs affecting mitochondrial function but leaving the pH of vacuole intact can trigger MDC formation and delivery? In other words, approach the generality of the system discovered upon mitochondrial dysfunction.

This point has been addressed in our response to all three other reviewers above. Again, we now include data showing that loss of mitochondrial membrane potential is not sufficient to activate this pathway in new Figure 3. This suggests that loss of vacuole acidity is necessary to activate this pathway through an undefined signal.

3) It is stated that the formation of small vesicle-like structures (Tom70 associated) in the cytoplasm of old cells require mitochondrial fission (Dnm1) indicating that these membrane structures are derived from mitochondria. It is very hard to see these structures in the picture of the dnm1 strain – perhaps a better version could be added? Also, as the experiment uses Vph1 as a reporter, does not this indicate that the vesicle-like structures are derived from the vacuole (vacuolar fragmentation?) rather than formed by fission from mitochondria? At present, it is a bit unclear what these structures are, how they were formed, and how they relate to the MDC route discovered.

We agree with the reviewer that these structures are hard to see in the originally submitted manuscript. As addressed in response to reviewer 3’s comments above, we have now included Video 1–Video 3, which show 3D reconstructions of old WT, *atg5△*, and *dnm1*△ cells taken with a DeltaVision microscope showing mitochondria (Tom70-GFP) and vacuole (Vph1-mCherry). As can be seen in the Video 1, the small Tom70-GFP punctae are outside of the fragmented vacuoles. In regards to the reviewer’s question about the origin of the fragments (mitochondria or vacuole), we believe the reviewer may have gotten confused with the labels here. The fragments are labeled with Tom70-GFP, not Vph1, and are thus mitochondrial derived. Vph1-mCherry is included in the pictures, but marks vacuoles.

4) The authors suggest that MDC formation helps protect mitochondria from vacuole-induced stress as a greater loss in membrane potential is seen in tom70/71 mutants subjected to conc-A treatment. Is it possible that this loss of potential in tom70/71 mutants could be related to other shortcomings than the formation of MDC?

Yes, we agree with the reviewer that the loss of membrane potential in the tom70/71 mutants could be unrelated to their role in the MDC pathway. We have now included a sentence in the manuscript addressing this alternative possibility.

Also, did atg5 and dnm1, but not atg32, mutants display a similar loss of potential upon conc-A treatment?

At the reviewer’s suggestion, we tested the membrane potential in these three mutant strains in untreated and conc A treated cells. We conducted this experiment by staining these cells with the membrane potential dye DioC6, and analyzing them by flow cytometry exactly as carried out in the paper. However, we found that all three of these mutants had membrane potentials lower than the WT strain even in untreated cells, which made the concA changes uninterpretable. Therefore, we did not include this data in the paper and instead added the qualifying statement mentioned in the point above.

5) Since aging is the entry port in this paper, I wonder if the authors could discuss briefly the possible significance of MDC in yeast longevity? As dnm1 and fis1 deletion mutants display a somewhat longer replicative lifespan, is it possible that a reduction in mitochondrial fragmentation (positive effect) in these mutants offsets the negative effect of an abrogated MDC response? Is the lifespan on tom70, atg5, atg32 mutants known?

Following the reviewer’s suggestion, we have added a brief discussion of the lifespan point regarding the fission mutants and MDC in the discussion of the paper. Because *dnm1△* mutants have an increased lifespan, we raised the possibility that the Dnm1-dependent mitochondrial fragments that form in old cells could negatively impact lifespan. The lifespan of *tom70*△ and *atg32*△ mutants has been shown to be longer than WT in dietary restriction conditions, but not necessarily normal growth conditions (Schleit et al., Aging Cell 2013). Lifespan data on *atg5△* has not been published. However, two other strains lacking genes required for general autophagy, *atg1△* and *atg8△,* have been shown to have an extended lifespan (McCormick MA et al., Cell Metab 2015). Based on all of this data, it is very hard to say whether the MDC pathway has any influence on lifespan. It may be the case the formation of the MDC is protective, and it doesn’t necessarily need to be degraded for protection.